

# A novel method to detect the tropopause structure based on bi-Gaussian function

Kun Zhang[1,2], Tao Luo[1,2], Xuebin Li[1,2], Shengcheng Cui[1,2], Ningquan Weng[1,2], Yinbo Huang[1,2], Yingjian Wang[1,2]

[1]Key Laboratory of Atmospheric Optics, Anhui Institute of Optics and Fine Mechanics, Hefei Institutes of Physical Science, Chinese Academy of Sciences, Hefei, 230031, China
[2]Advanced Laser Technology Laboratory of Anhui Province, Hefei, 230037, China

*Correspondence to*: Tao Luo (luotao@aiofm.ac.cn); Xuebin Li (xbli@aiofm.ac.cn)

**Abstract.** The tropopause is important as a diagnostic of the upper troposphere and lower stratosphere structures, with unique

atmospheric thermal, dynamic structures. A comprehensive understanding of the evolution of the fine tropopause structures is necessary and primary to further study the complex multi-scale atmospheric physicochemical coupling processes in the upper troposphere and lower stratosphere. Utilizing the bi-Gaussian function, a novel method is capable of identifying the characteristic parameters of tropopause vertical structures, as well as providing the information of double tropopauses (DT) structures. The new method improves the definition of cold point tropopause, and detects one (or two) most significant local

coldest point(s) in mathematical statistics by fitting the temperature profiles to the bi-Gaussian function, which is (are) defined as the tropopause height(s). The bi-Gaussian function exhibits remarkable potential for explicating the variation trend of temperature profiles. The recognition results of the bi-Gaussian method and lapse rate tropopause, as defined by World Meteorological Organization, are compared in detail for different cases. Results indicate that the bi-Gaussian method possesses a lower missed detection rate and false detection rate than lapse rate tropopause, because it is not restricted by thresholds, even

in the presence of multiple temperature inversion layers at higher elevations.  Five-year (from 2012 to 2016) historical radiosondes in China revealed that the occurrence frequency and thickness of DT, as well as the single tropopause height, and the first and second DT height displayed significant meridional monotonic variations. The occurrence frequency (thickness) of DT increased from 2.93 % (2.61 km) to 72.45 % (6.84 km) in the latitude range [16 ˚N, 50 ˚N]. At mid-latitudes [30 ˚N, 40 ˚N], the meridional gradients of tropopause height were relatively large, essentially corresponding to the climatological

location of the subtropical jet and Tibetan Plateau.  The average DT thickness reported in this study is approximately 1−2 km thicker than that in previous studies, particularly in the mid-high latitudes [45 ˚N, 50 ˚N], which may be related to the different vertical resolution of temperature profiles provided by various data sources.  DT structure occurs most frequently and has the largest meridional gradient in the mid-latitudes, formatted by a combination of poleward advection in the low-latitude upper troposphere and equatorward advection in the high-latitude lower stratosphere. In addition, although DT is thick in winter, the

DT temperature difference is small, even the case of the first tropopause temperature is lower than the second tropopause temperature happens occasionally.



# 1 Introduction

As a pivotal transitional layer uniting the troposphere and stratosphere, there are complex multi-scale atmospheric physicochemical coupling processes in the tropopause layer, such as atmospheric radiation and dynamical processes (Fueglistaler et al., 2009; Gettelman et al., 2011). Tropopause performs an essential role in stratosphere-troposphere exchange (STE) (Yang and Lv, 2003), mainly manifesting in the transportation of atmospheric energy and masses to the stratosphere vis this "gate" and further advection and diffusion (Yang and Lv, 2004; Fueglistaler et al., 2009; Holton et al., 1995). The long-time variations of the tropopause thermal and dynamic structures are regarded as crucial indicators of climate change (Santer et al., 2003b; Santer et al., 2003a; Sausen and Santer, 2003; Xian and Fu, 2017). However, the comprehension of the fine tropopause structures in is still limited (Bian et al., 2020), leading to varying degrees of biases in the tropopause inversion layer characteristics analysis (Randel et al., 2007b; Wang et al., 2013), climate model simulations (Li et al., 2020; Sun et al., 2021; Tian et al., 2017; Xian and Fu, 2015; Maddox and Mullendore, 2018).

It is beneficial to understand the formation mechanisms of tropopause, formatted by the combination of tropospheric and stratospheric processes, and to further research STE processes by defining the tropopause from various perspectives (Gettelman and Forster, 2002; Bian, 2009; Chen et al., 2006). In early atmospheric models, the tropopause is characterized as a discontinuous interface featuring a sharp vertical gradient. Reed (1955) proposed the concept of dynamic tropopause and discovered the tropopause folding events. Later, the dynamic tropopause was defined based on the zero-order discontinuity of potential vorticity (Danielsen et al., 1987). Based on the vertical structure of atmospheric temperature and the characteristics of a sharp decrease in the temperature lapse rate, the World Meteorological Organization (WMO) defined the thermodynamic tropopause (WMO, 1957). From a chemical point of view, Bethan et al. (1996) analysed whether there was a saltus in the atmospheric tracer concentrations through the thermal and dynamic tropopause, and set a threshold of the vertical gradient of ozone mixing ratio to represent the ozone tropopause (Pan et al., 2004; Pan et al., 2014; Ma et al., 2022).

It is more reasonable to consider the tropical tropopause as a transition layer than a discontinuous interface (Highwood and Hoskins, 1998). Gettelman and Forster (2002) comprehensively considered both radiation and convection, and separated cold point tropopause (CPT) and potential temperature lapse minimum rate tropopause (LRM) as the upper and lower boundaries of tropical tropopause layer, respectively. A primary characteristic of the tropopause is the drastic alteration of the atmospheric static stability when crossing this transitional layer. To gauge the thickness of tropopause layer, atmospheric static stability parameter buoyancy frequency $N$ has been introduced, characterized by the vertical potential temperature gradient (Homeyer et al., 2010). In ideal models, it is generally accepted that the static stability undergoes a discrete jump from low value (unstable) in the troposphere to high value (stable) in the stratosphere (Birner, 2006; Gettelman and Wang, 2015; Bai et al., 2017).

The meridional distribution of the thermal tropopause height, ranging from tropical to subtropical latitudes, often exhibits a cliff-like decline, rather than a continuous variation (Randel et al., 2007a; Rieckh et al., 2014; Schmidt et al., 2004). Coincidentally, in the overlapping tropical and mid-high latitudes, adjacent to the subtropical jet (STJ), double tropopauses



(DT) are frequently formed (Randel et al., 2007a; Xian and Homeyer, 2019; Schmidt et al., 2006). Fluctuations in atmospheric temperature resulting from different monsoon circulation systems, such as the Asian summer monsoon and polar vortex, can cause abnormal changes of tropopause height and increase the possibility of DT formation(Randel et al., 2007a; Rieckh et al., 2014; Ravindrababu et al., 2020; Shangguan et al., 2019). Currently, some studies have focused on atmospheric stability and tracers STE processes associated with the DT events, revealing that DT can impact the maximum water vapor

levels and stratospheric hydration in the lower stratosphere, as well as ultimately ozone concentration, transported by convective overshooting (Randel et al., 2007a; Pan et al., 2004; Gamelin et al., 2022; Homeyer et al., 2014b; Homeyer et al., 2014a). DT has an important influence on the vertical distribution, transport, and diffusion of atmospheric compositions, with active STE, and is a non-negligible key stratification when considering any mid-latitude stratosphere-troposphere activities (Peevey et al., 2014; Parracho et al., 2014; Liu and Barnes, 2018).

In order to deeply understand the coupling processes and triggering mechanisms involved in the upper troposphere-lower stratosphere (UTLS), the evolution of the fine tropopause structures must be comprehensively understood. However, the results of the existing tropopause identification methods are quite different in some cases, and the formation mechanism and evolution process of DT and tropopause inversion layer are still subject to controversy. Therefore, it is imperative to find a reliable and highly universal method to identify the characteristic parameters of tropopause vertical structures (Bian et al., 2020; Tian et

al., 2017).

The objective of this study is to introduce a new method to identify the multiple characteristic parameters of tropopause vertical structure. The temperature profiles obtained from radiosondes have been utilized to be fitted by a bi-Gaussian function, which can not only identify the tropopause height and tropopause temperature, but also express the information of DT structure, such as the thickness, as well as effectively assisting characteristic analysis on tropopause inversion layer. The key aspects of

this work are outlined as follows: Sec. 2 presents an account of historical radiosondes used in the study, commonly utilized definitions of thermal tropopause in previous researches, as well as a thorough description of the new identification method based on the bi-Gaussian function in details. The feasibility analysis of the new method, and comparisons between the new and LRT method are highlighted in Sec. 3. In Sec. 4, a comprehensive discussion of the spatiotemporal characteristics of the tropopause structures in China based on this new bi-Gaussian method is provided. The reasons for the formation of the DT

structure in the mid-latitudes and its vertical structure characteristics are discussed in Sec. 5. Ultimately, conclusions are summarized in Sec. 6.





# 2 Data and methods

## 2.1 Radiosondes

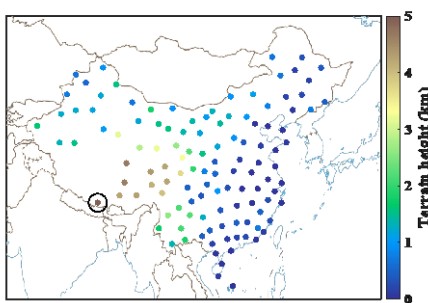

**Figure 1: Spatial distribution of sounding sites (dots) and relevant terrain heights (color-coded). The map is provided by the official website of the Ministry of Natural Resources of the People's Republic of China, NO: GS (2016) 1667 (the same as below).**

Despite there are numerous studies on the tropopause structures based on satellite data (Alexander et al., 2011; Liu et al., 2019; Rieckh et al., 2014), sounding observations play a crucial and essential role in directly obtaining basic atmospheric parameters in the UTLS, such as temperature and humidity profiles (Chen et al., 2006; Xian and Homeyer, 2019; Seidel et al.,

2001). The historical radiosondes used in this study were obtained from 120 sounding sites (color-coded in Fig. 1) in China from 2012 to 2016, as described in detail in the literature (Guo et al., 2016). Once- or twice- daily radiosondes throughout four seasons have a higher vertical resolution than reanalyses, and provide a wide meridional and zonal coverage, representing an excellent opportunity for a more precise identification of tropopause height, temperature, and DT structures. The basic information on the temperature profiles from radiosondes is listed in Table 1.

**Table 1: Information for high-resolution temperature profiles from radiosonde measurements.**

| Launch time | 08:00 and 20:00 (local time) |
|---|---|
| Sampling time | 1 s |
| Vertical resolution | 5 to 8 m |
| Measurement range | –90 to 50 °C |
| Temperature accuracy | ≤0.2 °C (–70 to 50 °C) |
| | ≤0.3 °C (–90 to 80 °C) |

## 2.2 Reanalysis

In order to investigate why DT structures in the mid-latitudes are frequent, ERA5 daily reanalysis are used to analyze the potential influence of atmospheric circulation anomalies on the tropopause vertical structure (Hersbach et al., 2020), including geopotential height, potential vorticity (PV, units: PVU, 1PVU=$10^{-6}$ m$^2$ K kg$^{-1}$ s$^{-1}$), temperature, V-component of wind (V),



vertical velocity (W), etc. The horizontal resolution of these parameters is 2.5 °× 2.5 °, and there are 37 vertical layers from

1000 hPa to 1 hPa.

The polar vortex intensity is defined as (Kolstad et al., 2010),

$$-Z_P = -\sum (Z' \cos \varphi) / \sum \cos \varphi \tag{1}$$

$$Z' = Z - \bar{Z} \tag{2}$$

Where, $Z$ and $\bar{Z}$ represent the 50 hPa geopotential height and its climatological mean, calculated during the period from January

1, 1961 to December 31, 2020, respectively. $\varphi$ is the latitude, and the summation symbols denote the sums of all grid points

north of 65 °N. $-Z_P$ denotes the polar vortex strength index, which is opposite to the sign of the geopotential height anomaly

in the polar region, as shown by the negative (positive) index corresponding to a weak (strong) polar vortex. The polar vortex

is an interaction between the upper (stratosphere) and lower (troposphere) parts, which influences the wind, pressure, and

temperature distributions in the Northern Hemisphere winter. This top-down circulation anomaly affects the winter climate

anomaly in East Asia, and the weak polar vortex is more easily transmitted from the stratosphere down to the troposphere and

have a significant effect on the tropospheric circulation, especially when the stratospheric outburst warming occurs in winter

(Chen and Wei, 2009).

**2.3 Previous thermal tropopause definitions**

In view of the fact that there is no physical meanings and highly universal definition of tropopause, the thermal tropopause

defined by WMO is widely utilized (Liu et al., 2019; Randel et al., 2007a; Rieckh et al., 2014; Schmidt et al., 2006; Schmidt

et al., 2004; Xian and Homeyer, 2019; Hoffmann and Spang, 2022; Feng et al., 2012). In addition, the CPT and LRM methods

are also adopted to characterize the tropopause boundaries (Seidel et al., 2001; Alappattu and Kunhikrishnan, 2010; Feng et

al., 2011; Zhang et al., 2022a). Next, the performance of these four thermal tropopause definitions was evaluated using the

radiosondes at a subtropical station (113.08ºE, 28.2ºN), as displayed in Fig. 2.

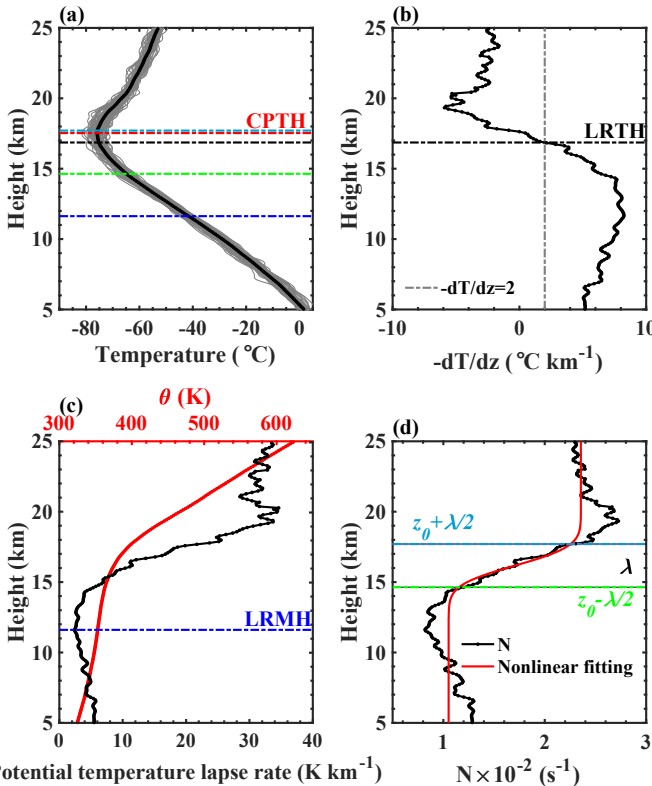

**Figure 2: Comparisons of tropopause height estimations utilizing four different thermal definitions based on radiosondes, a 15-point running mean was adopted for temperature profiles. (a) Comparisons of thermal tropopause height detected from different definitions. The grey lines are measured temperature profiles in January, 2014 at a subtropical station (113.08 °E, 28.2 °N), and the black dotted line denotes the average temperature profile of all grey lines. CPTH refers to the tropopause position corresponding the coldest point of the temperature profile. (b) LRTH signifies the lowest height above 500 hPa, where the temperature lapse rate is equal to or less than 2 ℃ km$^{-1}$ (WMO, 1957). (c) LRMH denotes the tropopause position corresponding the minimum of potential temperature lapse rate. (d) The tropopause upper ($z_0 + \lambda/2$) and lower ($z_0 - \lambda/2$) boundaries are determined by non-linear fitting to buoyancy frequency $N$.**

## 2.3.1 Tropical tropopause layer (TTL)

The atmospheric stability within the TTL is affected by convection in the troposphere and radiation in the stratosphere (Gettelman and Forster, 2002; Thuburn and Craig, 2002). Following further analysis and researches, two commonly used thermal tropopause definitions were emerged, namely CPT (in Fig. 2(a)) and LRM (in Fig. 2(c)). And the cold point tropopause height (CPTH) and lapse rate minimum height (LRMH) are defined as the upper and lower boundaries of TTL respectively, and their height difference is the TTL thickness (Gettelman and Forster, 2002; Alappattu and Kunhikrishnan, 2010). CPTH almost coincides with the minimum saturated water vapor mixing ratio (Fueglistaler et al., 2009; Kley et al., 1979; Randel and Park, 2019), and stratospheric water vapor concentration is mainly determined by the tropopause temperature (Rosenlof and



Reid, 2008; Rosenlof, 2003; Xie et al., 2020). Furthermore, LRMH holds three physical meanings (Gettelman and Forster, 2002; Ravindrababu et al., 2020):

1)    The maximum height at which convection still affects temperature in the upper troposphere;

2)    The height at which temperature begins to be influenced by stratospheric radiation;

3)    It coincides with the height corresponding to the minimum ozone mixing ratio.

### 2.3.2 Extratropical tropopause

The buoyancy frequency $N^2$ is an indicator of atmospheric static stability, which is characterized by vertical potential
temperature lapse rate (Homeyer et al., 2010; Birner, 2006; Gettelman and Wang, 2015; Bai et al., 2017), as following:

$$N^2 = \frac{g}{\theta}\frac{\partial \theta}{\partial z}, \tag{3}$$

Where, g is the gravitational constant, $\theta$ is the potential temperature, and $z$ is the altitude. A primary characteristic of tropopause layer is that the static stability changes drastically when crossing the tropopause layer. According to the discrete jump of $N$, $N$ is fitted by nonlinear least square method (in Fig. 2(d)) with the fitting function form

$$N(z) = N_{\text{trop}} + \frac{N_{\text{strat}} - N_{\text{trop}}}{2}\left[1 + \text{erf}(\frac{2(z-z_0)}{\lambda})\right], \tag{4}$$

Where, $N_{\text{trop}}$ and $N_{\text{strat}}$ are the asymptotic values of $N$ in the troposphere and stratosphere respectively, $z_0$ is the mid-point of the TH, $\lambda$ is the thickness of the transition layer, and erf represents the error function. $N_{\text{trop}}$, $N_{\text{strat}}$, $z_0$ and $\lambda$ can be obtained by nonlinear least squares fitting, with $N_{\text{trop}}$=0.0 s$^{-1}$, $N_{\text{strat}}$=25.0 s$^{-1}$ and $\lambda$ =1 km as the initial value of the fitting, and $z_0$ calculated according to the definition of WMO. In order to reduce the fitting error, data with altitude lower than 5 km are
discarded. Non-linear fitting of $N$ can directly obtain the upper and lower boundaries of the tropopause layer (in Fig. 2(d)).

Regrettably, the results of those existing methods are quite different in some cases (Wirth, 2000; Pan et al., 2018; Bian, 2009). It also can be seen from Fig. 2 that the results of the non-linear least squares fitting method are closer to CPTH, but quite far from LRMH at a subtropical site. Moreover, each definition is not universally employable, and most definitions hardly provide essential information about the transition layer structures.

### 2.4 New bi-Gaussian method

CPT and LRT are highly effective in the tropics, mainly due to the simple terrain and minimal impact of weather systems intrusion. But, the limitations of these two definitions become apparent in the extratropical, as shown in Fig. 3(a). The blue cross-line represents the measured temperature profile, and red dot-line is temperature lapse rate. According to the WMO algorithm, points C and D represent the first and second LRTH (LRTH1 and LRTH2), respectively. Point B (CPTH) is
quantitatively comparable with point D (LRTH2), however, CPT fails to identify point A, which corresponds to LRTH1 (point C). Therefore, CPT is unable to characterize DT structure.





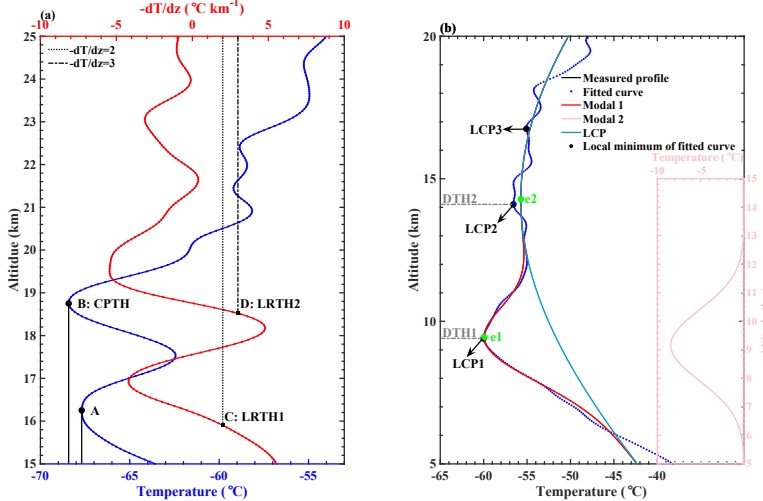

**Figure 3: (a) Schematic diagram illustrating the limitation of the CPT definition. The temperature profile was launched at a site situated over the Tibetan Plateau (TP) (87.08 ºE, 28.63 ºN, marked with a black circle in Fig. 1) @ 17:00, January 11, 2014 (Local Time, the same below). Points C and D are defined by the WMO: (1) The first tropopause is defined as the lowest level at which the lapse rate decreases to 2 ℃/km or less, provided that the average lapse rate between this level and all higher levels within 2 km does not exceed 2 ℃/km. (2) Above the first tropopause, if the average lapse rate between any level and all higher levels within 1 km exceeds 3 ℃/km, then a second tropopause is defined using the same criterion as under (1). (b) An example for bi-Gaussian function fitting to a temperature profile. The temperature profile launched at a site (119.7 ºE, 49.25 ºN) @ 19:14:25 at February 17, 2014. Above 14 km, modal 1 equals zero, and modal 2 perfectly coincides with the fitting curve. A 15-point running mean was adopted for temperature profiles.**

Inspired by Fig. 3, both points A and B represent local minimum points within a specific height range. Therefore, the local cold point (LCP) is used to replace the coldest temperature point in this study. The LCP is defined as follows: assuming that there exists a certain height $h_0$ (unit: km), where the temperature is the coldest in the interval $[h_0-1, h_0+1]$, we logically define $h_0$ as the local cold point height (LCPH), hereinafter referred to as "target". The target-seeking range $[TH_{min}, TH_{max}]$ is confined to minimize the identification error, corresponding to the bottom and top limits of TH, respectively (Liu et al., 2019), as following:

$$TH_{min} = 2.5 \times (3 + \cos(lat \times 2)),$$ (5)

$$TH_{max} = 2.5 \times (7 + \cos(lat \times 2)),$$ (6)

where, $lat$ is the latitude of observation sites.





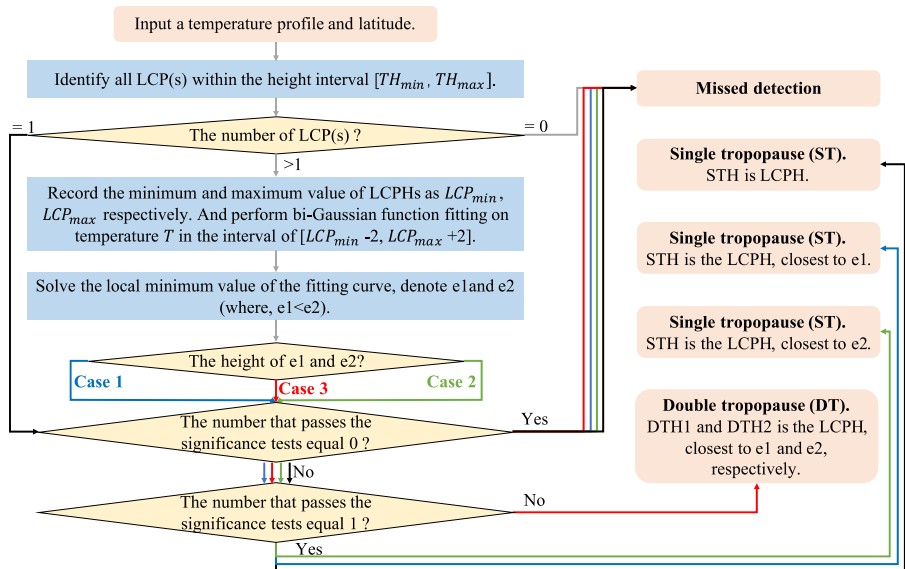

**Figure 4: The flow chart for utilizing the bi-Gaussian function to fit temperature profiles for identifying the tropopause height and the DT structure. The judgment criteria of Case 1, Case 2 and Case 3 are listed in Table 2.**

The detailed steps to identify the prominent target(s) are presented in Fig. 4, and the specific steps are delineated below.

1) **Target seeking**: Identify all LCP(s) within the height interval $[TH_{min}, TH_{max}]$. If the number of LCP(s) is equal to 1, proceed to step 4); otherwise, follow the subsequent steps.

2) **Function fitting**: Record the minimum and maximum values of LCPHs as $LCPH_{min}$ and $LCPH_{max}$, respectively. Bi-Gaussian function fitting is performed on the temperature profiles in the height interval $[LCPH_{min}-2, LCPH_{max}+2]$. The function is expressing:

$$f(x) = a1 * \exp\left(-\left(\frac{x-b1}{c1}\right)^2\right) + a2 * \exp\left(-\left(\frac{x-b2}{c2}\right)^2\right),$$  (7)

3) **Conditional judgment**: Solve the local minimum points of the fitted curve, namely $e_1$ and $e_2$, and judge whether $e_1$ and $e_2$ are within the interval of $[LCPH_{min}-2, LCPH_{max}+2]$, respectively. If the result is true, it is considered a valid value; otherwise, it is an invalid value.

4) **Significance tests**: Ensure that the average temperature lapse rate, which is represented by the slope of linear fitting to the temperature profiles in the range of [valid LCPH(s), valid LCPH(s)+1], is not less than 0.5 ℃/km (Guo et al., 2020; Randel et al., 2007b; Wang et al., 2013). Otherwise, it is invalid.

5) **Identification results**: Determine the LCPH(s) closest to the final valid value(s), which is (are) the tropopause height(s) (In the following, the abbreviation STH specifically refers to the single tropopause (ST) height, and DTH1 and DTH2 refers to the first and second DT height, respectively).

**Table 2: criteria in conditional judgments.**





| Conditional judgments | Criterion |
|---|---|
| Case 1 | $(LCPH_{min} - 2 \leq e_1 \leq LCPH_{max} + 2)$ is true & $(LCPH_{min} - 2 \leq e_2 \leq LCPH_{max} + 2)$ is false |
| Case 2 | $(LCPH_{min} - 2 \leq e_1 \leq LCPH_{max} + 2)$ is false & $(LCPH_{min} - 2 \leq e_2 \leq LCPH_{max} + 2)$ is true |
| Case 3 | $(LCPH_{min} - 2 \leq e_1 \leq LCPH_{max} + 2)$ is true & $(LCPH_{min} - 2 \leq e_2 \leq LCPH_{max} + 2)$ is true |

Fig. 3(b) shows an example of using the new method to identify THs, where h1 (LCP1) and h2 (LCP2) indicate the first DT height (DTH1) and the second DT height (DTH2), respectively. Table 3 summarizes the results and goodness of fit statistics from the bi-Gaussian function fitting to the temperature profile in Fig. 3(b).

**Table 3: The results and goodness of fit statistics of bi-Gaussian function fitting to the temperature profile in Fig. 3(b). The coefficient of determination $R^2 = 1 - \frac{SSE}{SST}$, the sum of squares due to error $SSE = \sum_{i=1}^{n}(X_i - Y_i)^2$, and $SST = \sum_{i=1}^{n}(Y_i - \bar{Y})^2$, where $X_i$ and $Y_i$ is the fitting and measurement value, respectively, and $n$ is the number of samples.**

| Fit parameters | | Goodness of fit statistics | |
|---|---|---|---|
| $a1$ | $-8.451$ $(-9.349, -7.552)$ | | |
| $b1$ | $9.158$ $(9.072, 9.244)$ | SSE | 55.4310 |
| $c1$ | $1.767$ $(1.609, 1.925)$ | | |
| $a2$ | $-55.72$ $(-55.82, -55.61)$ | | |
| $b2$ | $14.3$ $(14.07, 14.53)$ | $R^2$ | 0.9450 |
| $c2$ | $17.72$ $(16.53, 18.91)$ | | |

## 3 Feasibility analysis of the bi-Gaussian method

78,758 temperature profiles in 2014 were employed to discuss the feasibility of the bi-Gaussian method, including the capacity of the bi-Gaussian function to effectively interpret the temperature profiles, and comparing its advantages and limitations to LRT, which is widely applied to identify DT.





## 3.1 Explanation capability of bi-Gaussian function for temperature profiles

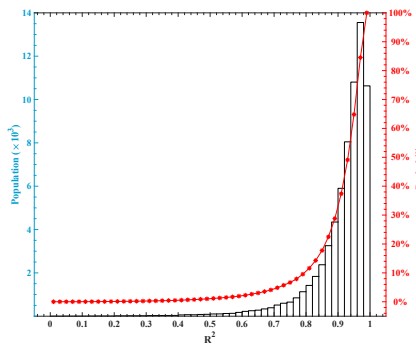

**Figure 5: Goodness of fit statistics (the coefficient of determination $R^2$) for bi-Gaussian function fitting to temperature profiles and cumulative probability distribution.**

Of the 78,758 temperature profiles, there are 68,896 profiles with no less than one LCP. Figure 5 shows the coefficient of determination $R^2$, one of the fitting evaluation indexes, of the 68,896 temperature profiles fitted by bi-Gaussian function. Higher $R^2$ values indicate better goodness of the regression model. $R^2$ is greater than 0.8 in at least 90% temperature profiles, and the average $R^2$ of all profiles reaches 0.9. Consequently, the bi-Gaussian function exhibits remarkable potential for accurately explicating temperature profiles.

**3.2 Comparisons of the identification results between the bi-Gaussian method and LRT**

**Table 4: Identification results of the bi-Gaussian method and LRT. The numbers and decimals in parentheses indicate the number of temperature profiles and their average $R^2$.**

| LRT ＼ Bi-Gaussian | Missed detection | ST | DT | Total |
|---|---|---|---|---|
| Missed detection | 384 (0.9) | 5,463 (0.88) | 1,702 (0.92) | 7,549 (0.88) |
| ST | 679 (0.91) | 45,850 (0.92) | **11,384 (0.92)** | 57,913 (0.92) |
| DT | 37 (0.89) | **4,536 (0.9)** | 8,723 (0.91) | 13,296 (0.91) |
| Total | 1,100 (0.91) | 55,849 (0.91) | 21,809 (0.92) | 78,758 (0.91) |

The tropopause structures of 78,758 profiles, identified using two methods, are summarized in Table 4. The bi-Gaussian method reveals that 1.4 % temperature profiles' LCP(s) are either beyond the range of $[TH_{min}, TH_{max}]$ or do not meet the
significance tests, thereby causing missed detection. Because LRT are limited by thresholds, 9.59 % temperature profiles fail to satisfy the criterion of "the average lapse rate between this level and all higher levels within 2 km does not exceed 2 ℃/km", leading to missed detection. It is obvious that the bi-Gaussian method has a lower missed detection rate than LRT. Similarly, some temperature profiles are unable to meet the criterion that "the average lapse rate between any level and all higher levels




within 1 km exceeds 3 ℃/km", and thus, LRT identifies them as a single tropopause structure, resulting in the bi-Gaussian
method detecting 10.81 % more DT than LRT.

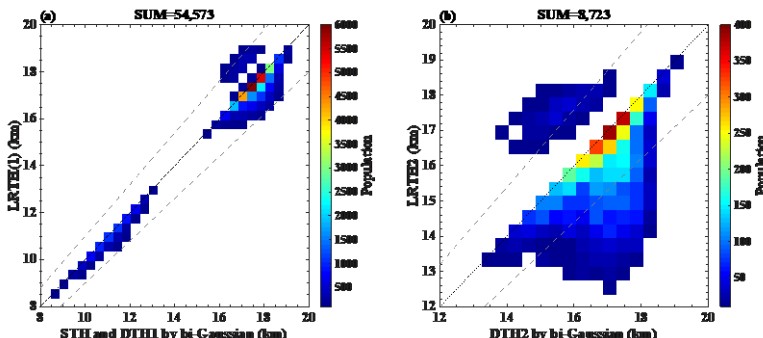

**Figure 6: Comparison of identify results between the new bi-Gaussian method and LRT, the fill colormap represents the number of points, and the gray dotted lines indicate the 10 % error threshold; (a) STH and DTH1; (b) DTH2**

Figure 6 shows the comparison of the identification results obtained through the new bi-Gaussian method and LRT. Both
methods can identify 69.78 % cases with the same structural types. Specifically, the correlation coefficients of the two methods
for the STH and DTH1 are 0.91 (in Fig. 6(a)), and at least 91.92 % (98 %) of points' error between the two methods is no more
than 10 % (30 %). Although the second DT height identified by the two methods are characterized by a more dispersed
distribution (in Fig. 6(b)), 67.47 % (97.87 %) of points have a percentage difference of no more than 10 % (30 %), with the
darkest patches are located on the line y=x. Further, TH2 identified by the bi-Gaussian are general higher than LRTH2, and
the average $R^2$ of 2,838 profiles with a percentage difference of more than 10 % is 0.91, of which the $R^2$ of the 2,604 (91.75 %)
profiles are not less than 0.8.

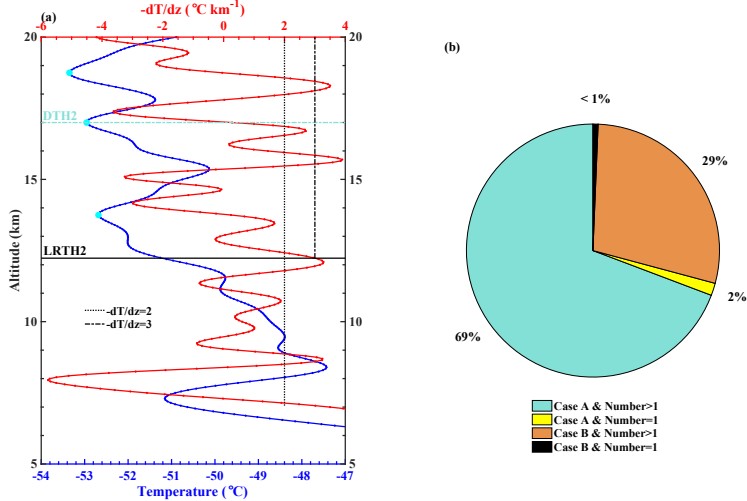

**Figure 7: The characteristics of the 2,838 temperature profiles, which hold a difference of more than 10 % in the second DT height between the bi-Gaussian and LRT. (a) A typical profile to illustrate the possible reasons for the difference. The temperature profile**
**was launched at the site (123.9167 ºE, 47.3833 ºN) @ 07:31:25, November 12, 2014; (b) According to the number of inversion layer(s) in the height range of [10 km, 20 km] and whether there is a higher and colder LCP, the 2,838 profiles are classified into four distinct**




clusters. These clusters are deliberated for the purpose of gathering characteristic statistical information. (Case A indicates that presents of a higher and colder LCP, and Case B is the converse of Case A).

Figure 7 shows the characteristic statistical analysis of the 2,838 temperature profiles with significantly different second

DT height (a percentage difference of more than 10 %) between the bi-Gaussian and LRT. A typical temperature profile (in

Fig. 7(a)) is exampled to illustrate the possible reasons for the difference in the second DT height between the two methods.

This typical profile displays two obvious characteristics. Firstly, three inversion layers are formed in the height range of [10

km, 20 km], meaning that there may be no less than one point that fit the criteria of the second tropopause height as defined

by WMO. Consequently, LRTH2 may be ambiguous due to the existence of multiple inversion layers (Hoerling et al., 1991).

Secondly, it is evident that at least one colder LCP above the first tropopause, frequently accompanied by a more significant

temperature inversion layer. Notably, the temperature of all three LCPs (cyan dots) above the first tropopause height are lower

than the first tropopause temperature. Out of the 2,838 profiles examined, more than 99 % profiles possess at least one of the

features (in Fig. 7 (b)), and even 69 % possess both features simultaneously. Among them, more than 97 % profiles exhibit

multiple inversion layers in the height range of [10 km, 20 km]. Similar temperature profiles reveal the disadvantage of LRT

constrained by a threshold will be exposed, which is an important superiority of the bi-Gaussian method over LRT.

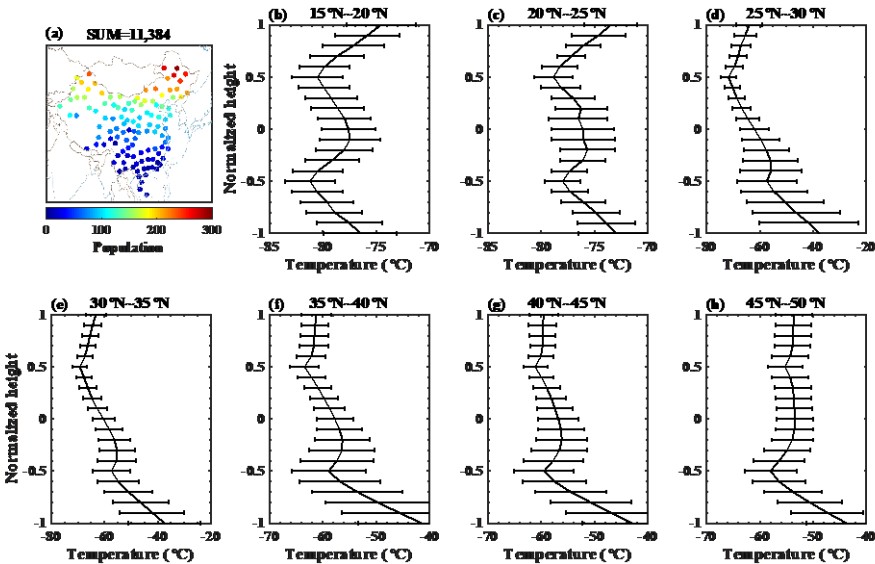

**Figure 8: The cases (the red marked in Table 4) that were identified as DT by the bi-Gaussian method but ST by LRT. (a) Sites distribution of the 11,384 temperature profiles. (b) − (h) annual average temperature profiles in 5° latitude bands. A tropopause-based average is adopted by using (DTH1+ DTH2)/2 as a reference level, with the vertical coordinates derived accordingly, and −0.5**

**and 0.5 accurately indicate the locations of DTH1 and DTH2, respectively.**

There are 23,801 temperature profiles with contradictory results from the LRT and bi-Gaussian method, of which the

largest proportion (11,384 profiles) is identified as DT by bi-Gaussian method, but ST by LRT. Fig. 8 shows the sites

distribution of the 11,384 temperature profiles (bolded in Table 4) and the annual average temperature profiles with normalized



height in each latitude zone. The average $R^2$ of the bi-Gaussian function fitting for these11,384 temperature profiles is 0.92,
of which the $R^2$ of the 10,588 profiles is not less than 0.8. Using (DTH1+ DTH2)/2 as a reference level, the tropopause-based
annual average temperature profiles of each latitude zone (in the interval of 5°) are calculated, as shown in Fig. 8(b) − (h).
There is an obvious peak at −0.5 on the annual average temperature profiles in all latitudes, but the bimodality become
gradually unobvious poleward, especially at the peak at 0.5. LRT is too insensitive to detect weak inversion signals, which
may be the reason for the large proportion of contradictory results in the mid-high latitudes.

Similarly, there are 4,536 temperature profiles (bolded in Table 4, not shown) being identified as DT and ST by the LRT
and new method, respectively. The main explanation for this phenomenon is that, there is singular LCP, reflecting a more
significant inversion layer, is present on the bi-Gaussian function fitting curve within the range of $[TH_{min}, TH_{max}]$. But the
average $R^2$ of those profiles is 0.9, and $R^2$ of 4,016 profiles is not less than 0.8.

## 4 Spatiotemporal characteristics of tropopause structures in China

### 4.1 Occurrence frequency and thickness of DT

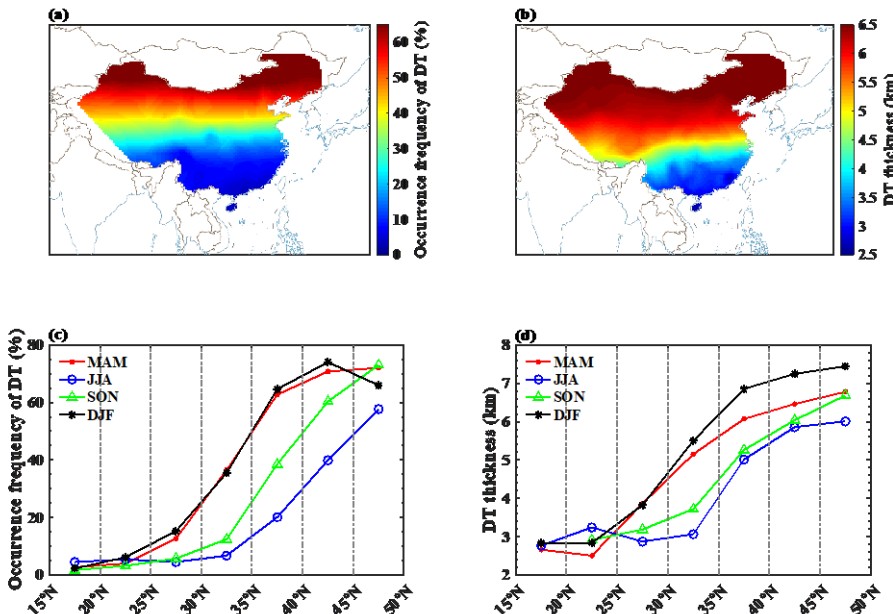

**Figure 9: The meridional distribution of the annual (a, b), seasonal (c, d) mean occurrence frequency (a, c), and thickness (b, d) of DT in China during 2012−2016. Zonal means were determined for 5°latitude bins, and the same below.**

In general, the occurrence frequency in Fig. 9(a) (thickness in Fig. 9(b)) of DT reaches its maximum in the Northern
China and gradually decreases towards the tropics, showing typical meridional distribution characteristics (Schmidt et al., 2006;
Seidel and Randel, 2006; Johnston and Xie, 2020). The maximum of annual mean occurrence frequency (thickness) is about





72.45 % (6.84 km), and the minimum is about 2.93 % (2.61 km) in the range [16 ºN, 50 ºN]. Additionally, the annual mean occurrence frequency in the mid-latitude area [30 ºN, 45 ºN] is about 43.53 %, which is roughly 6.09 times the occurrence frequency (7.15 %) in the low-latitude area [20 ºN, 30 ºN]. The phenomenon that DT occurs frequently in mid-latitudes has

also been recorded in previous studies and explained as the influence of the subtropical jet (STJ) system, whose mean climatological distribution position is about [30º, 45º] (Schmidt et al., 2006; Seidel and Randel, 2006), and the susceptibility of tropopause folding events in STJ region (Elbern et al., 1998). The thickness in the area [90 ºE−100 ºE, 26 ºN−32 ºN] is obviously greater than that of the same latitudinal plain, which may be affected by the giant topography of the TP.

Fig. 9(c) shows the zonal mean occurrence frequency of DT, determined for 5 °latitude bins, in spring (March-April-May,

MAM), summer (June-July-August, JJA), autumn (September-October-November, SON) and winter (December-January-February, DJF). Taking 30 ºN as the dividing line, the occurrence frequency increases sharply from low to middle latitudes, particularly in winter and spring. DT occurrence frequency is not only low (<10 %) but also no significant seasonal variations in low latitudes. However, it shows obvious seasonal differences in middle latitudes, reaching the largest (~35.52 %−74.03 %) and the smallest (~6.73 %−57.71 %) in winter and summer, respectively. The mean climatological location of STJ gradually

moves southward from [40 ºN, 45 ºN] in summer to [30 ºN, 35 ºN] in winter (Holton, 2004), and concurrently becomes stronger. In consequence, a large meridional gradient of DT occurrence frequency usually occurs in the climatological location of STJ and its adjacent latitude zone (±5º). DT occurrence frequency in the mid-high latitudes [45 ºN, 50 ºN] does not maintain an increasing trend in winter, showing a downward trend, which is consistent with the result in (Randel et al., 2007a; Schmidt et al., 2006). However, such downward trends are not observed during other seasons.

In Fig. 9(b) and Fig. 9(d), the annual and seasonal mean DT thickness in mid-latitude regions [30 ºN, 50 ºN] are different from previous study (Schmidt et al., 2006), representing a numerically higher ~1 km (~2 km) in winter (in summer) and a slight increasing trend. The atmospheric temperature in UTLS over the northern China, especially in the mid-high latitudes, is primarily affected by cold air intrusion caused by weather systems, such as the Siberian High, the polar vortex, and the Asian winter monsoon (He and Wang, 2016; Woo et al., 2015; Shangguan et al., 2019). Strengthened north winds are conducive to

the advection and subsidence of cold air into East Asia, resulting in the hot and cold currents converge in the upper troposphere, which is beneficial to form significant temperature inversion layers. Bi-Gaussian method prefers to define higher and colder temperature inversion layer as DTH2, which leads to an increase in the occurrence frequency and thickness of DT.





## 4.2 Tropopause heights

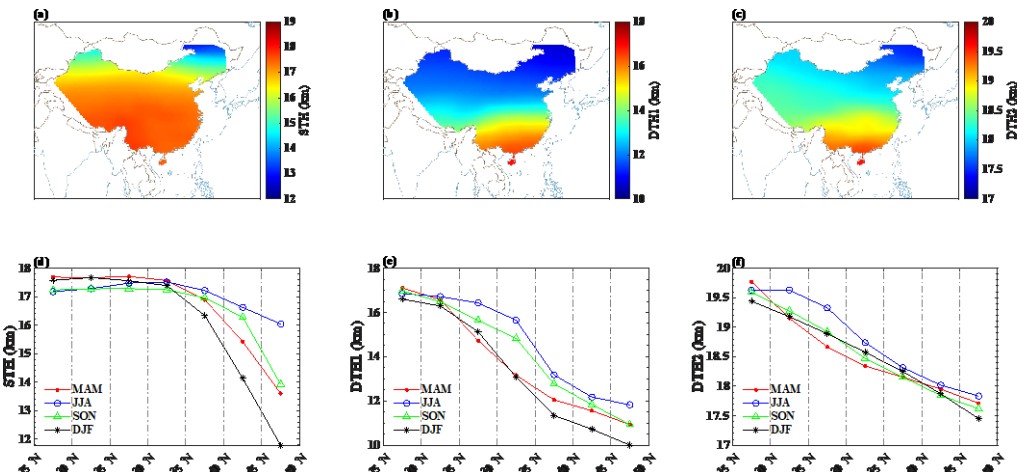

**Figure 10: The meridional distribution of the annual and seasonal mean tropopause height in China during 2012−2016. The first row corresponds to the annual mean STH (a), DTH1 (b), and DTH2 (c). The second row represents the seasonal STH (d), DTH1 (e), and DTH2 (f) in spring, summer, autumn, and winter, respectively.**

As shown in Fig. 10, STH, DTH1 and DTH2 also manifest a significant meridional structure, which is similar to LRTH (Schmidt et al., 2004) and CPTH (Tang et al., 2017). STH (DTH1, DTH2) gradually decreased from 17.95 km (16.97 km, 19.61 km) to 12.64 km (10.61 km, 17.39 km) in the range [16 °N, 50 °N], with DTH2 appearing to display a more subdued fluctuation. Schmidt et al. (2004) revealed that the mean (first) LRTH of the margin of the tropics from 2001 to 2003 was approximately 15.8 km, slightly lower than the result (16.88 km) determined by the bi-Gaussian method. There are two potential reasons accounting for this deviation. Firstly, this may be determined by the inherent properties of the two methods. For the same inversion layer, bi-Gaussian focuses on the location of LCP (the lowest temperature point of an inversion layer), while LRT's search range is restricted to where the temperature decreases with height, resulting in the recognition results of bi-Gaussian being inevitably higher. This factor may also account for the phenomenon that LRTH universally lies below the CPTH as reported in earlier studies (Gettelman and Forster, 2002; Schmidt et al., 2004; Kim and Son, 2012), such as the individual case in Fig. 2. Secondly, it cannot be ignored that tropopause height has an increasing trend under global warming (Santer et al., 2003a; Tang et al., 2017).

The change trend of tropopause height from tropical to subtropical regions is discontinuous, characterized by a cliff-like fall, corresponding to the DT frequency zone (Randel et al., 2007a; Rieckh et al., 2014; Schmidt et al., 2004; Feng et al., 2012; Xian and Homeyer, 2019; Pan et al., 2004). The meridional gradients of STH, DTH1 and DTH2 in mid-latitudes [30 °N, 40 °N] are significantly larger (in Fig. 10(d)−(f)), which basically corresponds to the climatological location of STJ and TP. The tropopause structures in the mid-latitudes are asymmetrical in both hemispheres, with a greater complexity in the Northern hemisphere (Xia et al., 2021; Han et al., 2011). In the Northern hemisphere, the meridional gradient of the first tropopause



height is steeper (slower) in summer (winter) (Randel et al., 2007a), and DT occurs more frequently (Johnston and Xie, 2020; Schmidt et al., 2006; Zeng et al., 2017) than that in the Southern hemisphere. TP, an important source of gravity waves coupled with its very representative giant topography (Hoffmann et al., 2013; Khan et al., 2016), may be one of the contributors to this variance. During winter, the subtropical westerly jet is located on the southern margin of the TP (Chen et al., 2006). and

atmospheric fluctuations triggered by topography, jet-stream or convection (De La Torre et al., 2004), manifesting as strong variations in static stability (Koch et al., 2005), increase the probability of forming an inversion layer at a lower height. In summer, a strong Asian summer monsoon anticyclone is prevailing under the thermal difference between sea and land (Xu et al., 2019; Park et al., 2009; Bian et al., 2020; Liu et al., 2017; Wu et al., 2016; Ma et al., 2023), which can destabilize atmospheric temperature stratification in UTLS by inspiring deep convection and monsoon circulations (Randel and Park,

2006). However, the contribution of the unique thermal and dynamic effects of the TP in different seasons to the tropopause structures in local and surrounding regions need further study.

## 5 Discussion

The conceptual model of tropical STE summarized by Holton et al. (1995) is now widely recognized. However, the extratropical mid-latitudes, with the sharp meridional gradients in the tropopause construes, have a complex atmospheric

dynamic processes stratospheric dynamical processes (Butchart, 2022), like the stratospheric polar vortex. Therefore, the stratosphere-troposphere coupling process and its conceptual model still need to be further studied (Huang et al., 2018). Previous studies (Kolstad et al., 2010; Tomassini et al., 2012) have identified an association between temperature anomalies in the Northern Hemisphere and the strength of stratospheric polar westerlies (Christiansen, 2001; Zhuo et al., 2022). Taking winter season as an example, we will discuss the analysis of potential causes contributing to frequent DT structures in the mid-

latitudes.

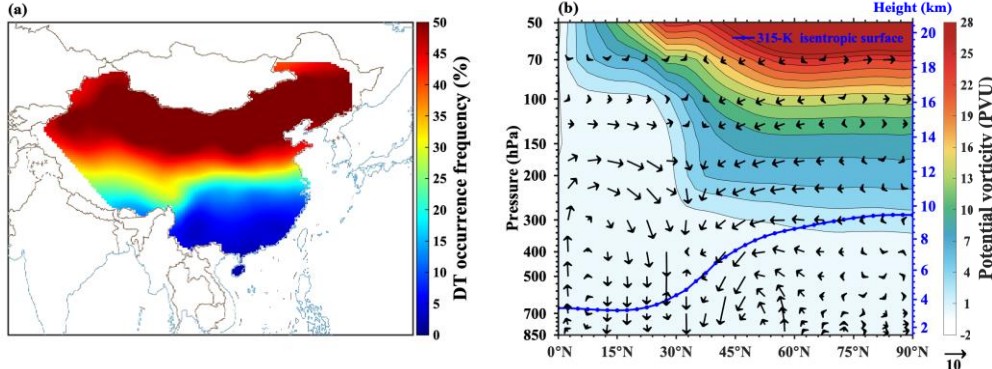

**Figure 11: (a) Characteristics of the spatial distribution of DT occurrence frequency in China during the winter (DJF) from 2012 to 2016 based on radiosondes and the new Bi-Gaussian method; (b) Latitude–height cross section of PV (shading) and wind field (vectors, vertical velocity is scaled by –500) along the 75 °E–130 °E, and the blue dot line indicates the locations of the 315 K isentropic**

**surface at different latitudes.**



During the winter (DJF) from 2012 to 2016, the average DT occurrence frequency in the range of [30 °N, 50 °N] (as shown in Fig. 11(a)) was about 46.33 %, decreasing sequentially from north to south. The maximum is concentrated in the latitudinal band of [40 °N, 46 °N], and decreases sharply in the latitudinal band of [30 °N, 35 °N], and the DT occurrence frequency south of 30 °N is basically less than 10 %. This trend is consistent with previously reported results (Bartusek et al.,

2023; Lin et al., 2023; Liu and Barnes, 2018). As can be clearly seen from Fig. 11(b), significant poleward (equatorward) meridional advection exists south (north) of the mid-latitudes in upper troposphere (lower stratosphere) at low latitudes (high latitudes). The high PV (>4.5 PVU) and high static stability air mass, makes equatorial advection on the isentropic surfaces (such as the 315 K isentropic surface, blue dot line in Fig. 11(b)) and invades over the mid-latitudes, called the "polar waveguide" (Zhang et al., 2022b; Zhang et al., 2023). Upper tropospheric low PV (<2 PVU) and low static stability air mass

carried by the polar advection, advection to the middle latitudes above, called "low latitude waveguide". The north and south waveguides are advected at 100hPa and 300hPa layer respectively, causing PV anomalies that PV contour bows up and bows down at the high altitude of 30 °N. As a result, the mid-latitudes act as a transition zone between warm and cold air masses, exhibiting discontinuities in temperature gradients. In addition, the 315 K isentropic surface has the largest downward tilt rate from 60 °N to 30 °N, corresponding to the region with the highest and the largest meridional gradient of the DT occurrence

frequency, which indicates that the advection motions are closely related to the formation of DT structure.

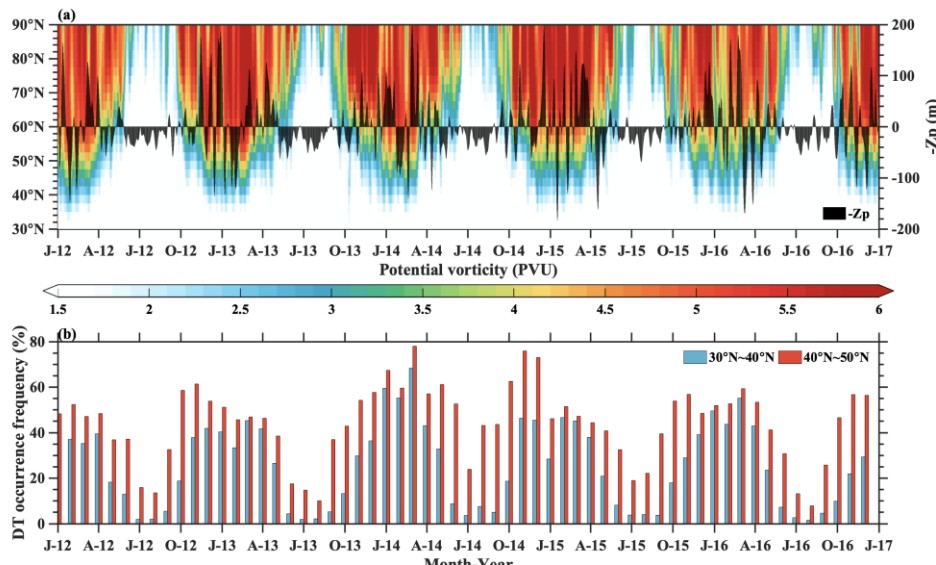

**Figure 12: (a) Latitude–time cross section of the monthly mean potential vorticity (shading) and daily $-Z_P$ (black bar; units: m) averaged along 75 °E–130 °E from January 2012 to December 2016; (b) the monthly mean DT occurrence frequency at the range of [30 °N, 40 °N] and [40 °N, 50 °N] based on radiosondes (along 75 °E–130 °E) from January 2012 to December 2016.**

Fig. 12(a) shows the latitude–time cross section of the monthly mean PV at 315 K isentropic surface and daily polar vortex intensity $-Z_P$ from January 2012 to December 2016. Both the peaks and valleys of $-Z_P$ occur in winter, indicating the





frequent and alternating occurrence of strong and weak polar vorticity. The tropopause height in the polar and high-latitudes is lower than that in the mid- and low-latitudes, so the height of the lower stratosphere (with a high PV) in the polar regions and high latitudes corresponds to the upper troposphere (with a low PV) in the mid-latitudes. Aforementioned trend, Fig. 12(a)

vividly illustrates the southward movement (northward contraction, i.e., northward movement of tropical air masses) of polar (tropical) air masses in winter (summer). In winter, extremely cold air with high PV (>4.5 PVU) driven by polar vortex activities, invades southward along the sloping 315 K isentropic surface (Zhang et al., 2022b; Zhang et al., 2023), even the 2 PVU contour line even crosses the 40 °N. There was no similar pattern in summer.

The latitudinal band of [30 °N, 50 °N] is the tropopause break area, as well as the intersection area of polar and tropical

air masses, which may be one of the reasons for the frequent DT structures in the mid-latitudes. In winter, the DT occurrence frequency in the [30 °N, 50 °N] latitudinal band reached 47.87 %, with the DT occurrence frequency in the [40 °N, 50 °N] latitudinal band being 54.42 %, which was higher than that in the [30 °N, 40 °N] latitudinal band (41.3 %). The higher DT occurrence frequency in the [40 °N, 50 °N] reinforces the impact of stratospheric polar vortex intrusion at high latitudes.

In the Northern Hemisphere, the stratospheric processes in the mid- and high-latitude during winter mostly are

characterized by downward propagation to the lower troposphere (Christiansen, 2001), in which upper-level PV anomalies can vertically influence lower-level regions by regulating meridional circulations (Black, 2002). PV anomalies in the upper troposphere favor the southward movement of the accumulated cold Siberian air into the mid-latitudes, and then downward transport, disrupting the vertical structure of atmospheric temperature and causing the discontinuities of temperature gradient. How does the above atmospheric circulation anomalies affect the characteristic parameters of DT vertical structures?





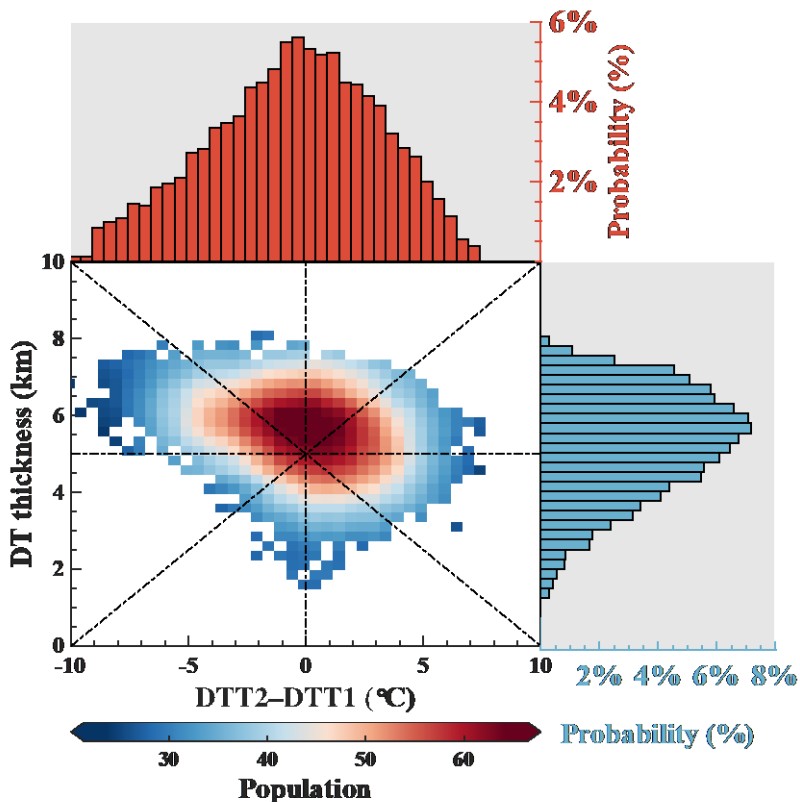

**Figure 13: Frequency distribution between the DT temperature difference ΔT (= DTT2–DTT1, DTT1 and DTT2 denotes the first tropopause temperature and the second tropopause temperature, respectively) and the corresponding DT thickness for the [30 °N, 50 °N] latitude band (heatmap in the center). The upper and right histograms show the probability distribution of the ΔT, and DT thickness, respectively.**

**Table 5: Statistics of tropopause heights and temperatures in the latitudinal band of [30 °N, 50 °N] during the winter from 2012 to 2016.**

|  | Tropopause height (km) | | | Tropopause temperature (°C) | | |
|---|---|---|---|---|---|---|
|  | STH | DTH1 | DTH2 | STT | DTT1 | DTT2 |
| Average | 14.09 | 10.87 | 16.33 | –64.3 | –59.81 | –61.44 |
| Median | 15.41 | 10.76 | 16.53 | –64.4 | –60.03 | –61.51 |

Fig. 13 and Table 5 illustrate the statistical characteristics of the tropopause vertical structures (tropopause height, tropopause temperature and DT thickness) in the latitudinal band of [30 °N, 50 °N] during the winters from 2012 to 2016.The DT thickness is greater than 5 km in more than 63% of the cases, which indicates the deepness of the DT structure. ΔT (= DTT2–DTT1) and DT thickness are concentrated at [–3 °C, 3 °C] and [4 km, 7 km], and the mean values are –1.64 °C and



5.45 km, respectively. This means that even if there is a difference of 5.45 km between DTH2 and DTH1, DTT1 is lower than DTT2. And the frequency of cases where DTT1 is lower than DTT2 is about 57 %, which is more than half of the cases. The stratospheric high PV at high latitudes induces the stratospheric advection invasion of cold air from Siberia into the upper mid-latitude troposphere (Tomassini et al., 2012),especially when the inverted omega-shaped circulation pattern is formed, cold
spells are very prone to break out (Zhang et al., 2022b; Zhang et al., 2023). The convergence of warm and cold air not only forms a prominent thermal inversion layer at the first tropopause, but also drastically reduces the local minimum temperature, and even a few cases occur with TT1<–65 ℃ (ΔT< –8 ℃).

Why is DTH2 elevated and is DTT2 slightly larger than STT? The poleward warm advection in the upper troposphere at low–latitudes heats the upper atmospheric temperature in the mid-latitudes, increasing buoyancy and making an upward motion,
which elevates DTH2 and increasing DTT2. Anticyclonic vorticity is generated above the heat source, which is realized as negative PV anomaly. Combined with the positive PV anomaly, due to the advection invasion of equatorward high PV, this may increase the static instability within the deep tropopause layer, and exacerbate atmospheric mixing and STE processes (Liu and Barnes, 2018).

## 6 Conclusions

This study presents a reliable and highly universal method for identifying tropopause structures, based on the concept of LCP. In this method, temperature profiles are fitted with the bi-Gaussian function to find the most significant one (two) LCP(s) in mathematical statistics, which is (are) regarded as tropopause height(s). The bi-Gaussian function achieves a good explanatory potential for temperature profiles, with an average coefficient of determination $R^2$ reaches 0.9, of which more than 90 % temperature profiles exhibiting $R^2$ values are greater than 0.8. The recognition results of the bi-Gaussian method
and LRT are compared in detail for different cases. Firstly, 69.78 % temperature profiles were identified as the same structure by the two methods with significant correlation (the correlation coefficient of the single and first tropopause heights is 0.91). Secondly, contradicting or slightly different results are mainly found in the mid-high latitudes. The differences may be related to the temperature fluctuations in the UTLS caused by weather systems, especially multiple higher and colder inversion layer(s) formed, indicating the ambiguity of LRT constrained by thresholds. Nonetheless, the average $R^2$ of those profiles is not less
than 0.9. In general, the bi-Gaussian method possesses a notably lower rate of missed and false detection, because of being free from the limitation by any threshold. The bi-Gaussian method is able to effectively express the DT structural information, offering a prominent advantage over CPT and LRM. Of course, the new method can also assist the researches on tropopause inversion layer characteristics.

Five-year (from 2012 to 2016) historical radiosondes in China showed that tropopause structures (occurrence frequency
and thickness of DT, STH, DTH1, and DTH2) displayed significant monotonous meridional distribution characteristics. In the latitude range of [15 ºN, 50 ºN], STH (DTH1, DTH2) gradually decreases from 17.95 km (16.97 km, 19.61 km) to 12.64 km (10.61 km, 17.39 km), and DT occurrence frequency (thickness) increased from 2.93 % (2.61 km) to 72.45 % (6.84 km), with



a steep variation in middle latitudes. Subtropical regions [15 ºN, 25 ºN] exhibit ST-dominated conditions throughout the year, while mid-high latitudes [15 ºN, 35 ºN] experience high frequency of DT occurrence, particularly in winter where the
occurrence frequency exceeds 70 %. This may be related to STJ activities and the intrusion of cold air from the north, caused by weather systems in winter. Notably, the climatic location of STJ and its adjacent latitude zone (±5º) exhibits a sharp increase in occurrence frequency. Furthermore, the DT thickness in the mid-high latitudes during winter is not less than 5 km. The average DT thickness in this study is about 1−2 km thicker than previous results, especially in the mid-high latitudes [45 ºN, 50 ºN], which may be related to the different vertical resolution of temperature profiles provided by various data sources
(König et al., 2019; Hoffmann and Spang, 2022; Pan et al., 2004; Wang et al., 2019). Moreover, tropopause structures over the Tibetan Plateau differ from those in the same latitudinal zone, likely due to unique atmospheric circulation structures such as the Asian summer monsoon anticyclone, planetary wave breaking and uploading, orographic gravity waves, and atmospheric temperature disturbances in the UTLS. However, the underlying mechanisms require further investigation.

In conclusion, the conceptual model of DT formation in the mid–latitudes during the winter is summarized. The deep DT
structure is formatted by the combined action of significant poleward and equatorward meridional advection. Meridional advection, driven by the polar vortex, of high–PV cold air from the polar regions and high latitudes into the mid–latitudes forms DT1 while cooling DTT1. On the other hand, the poleward advection of low-latitude waveguide in the low–latitudes upper troposphere elevates the DTH2 while warming DTT2.

**Data availability.** The radiosonde data used in this study are available upon the reasonable request from the corresponding
author (luotao@aiofm.ac.cn).

**Author contributions.** KZ and TL jointly developed the concept of this study, and wrote the manuscript. XL prepared the radiosonde data sets. NW, YH, and YW conducted the data analysis, and contributed to the interpretation of the results. SC gave the financial support. All authors have read and agreed to the published version of the manuscript.

**Competing interests.** The authors declare that they have no conflict of interest.

**Acknowledgements.** This research has been supported by the Project funded by China Postdoctoral Science Foundation (Certificate Number: 2023M733537), the Youth Fund Project of Advanced Laser Technology Laboratory of Anhui Province (Grants no. AHL2022QN02, AHL2021QN01), the HFIPS Director's Fund (Grant No. YZJJ2023QN07), the Anhui Provincial Natural Science Foundation (Grant no. 2008085J19), and the Special Project of Nanhu Laser Laboratory (Grant no. 22-NHLL-ZZKY-005).

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
