# Peer review of "A novel method to detect the tropopause structure based on bi-Gaussian function"

_EGUsphere, 2024_

## Author Response (AR1)

Dear reviewers,

We feel great thanks for your professional review work on our manuscript. As you are concerned, there are several problems that need to be addressed. According to your nice suggestions, we have made extensive corrections to our previous draft, the detailed corrections are listed below. In addition, we have written a Supplement, listing the definitions of the two methods (Bi-Gaussian and lapse rate tropopause (LRT)) and a wide variety of scenarios that may be encountered in the process of identifying the tropopause structure in details, in order to demonstrate the differences of two different methods. We hope the Supplement will help you understand the results in the main text.

**RC1:**

**Comments:**

This work uses a novel method to detect tropopause especially the double tropopause and shows the statistics of the tropopause using this new method. The definition of the tropopause is an important question in the upper troposphere lower stratosphere dynamics and chemistry, and the information presented in this paper is very quantitively and clear. However, the discrepancy between this new method and existing method seems to be too large and I suggest the authors show more clarification in section 3, please see major comments for details. I suggest accepting this paper after resolving this issue.

**Major comments:**

1. Figure 8, and lines 281-290. This part needs more detailed discussion and clarification:

Line 281-282: "of which the largest proportion (11384 profiles) is identified as DT by bi-Gaussian method but ST by LRT", this result is astonishing. According to the WMO double tropopause definition, 'If the average lapse rate above this "first tropopause" between any level and all higher levels within 1 km exceeds 3°C/km, then a "second tropopause" is defined by the same criterion as the first', I checked the significance test

in section 2 and only find that "the slope is not less than 0.5°C/km", is it a reason for this very different result between the bi-Gaussian method and LRT?

**Reply:**

1) The threshold for "the slope" in the significance test does affect the identification of the second tropopause. The threshold of 0.5 °C/km in the manuscript is set based on the statistical results of a research (Fig.5 in Randel et al. (2007b)], the average strength of the inversion is defined by the temperature difference between $Z_{LRT1}+2$ km and $Z_{LRT1}$) (Randel et al., 2007b). In the revised manuscript, we also explained that the threshold of "the slope" is one of the reasons for the difference in recognition results between the two methods.

2) In order to avoid misjudgment of the tropopause structure due to local temperature fluctuations caused by atmospheric fluctuations, we have improved constraints in the significance test, by changing the range of the linear fitting to [valid LCPH(s), valid LCPH(s)+2] (referring to Randel et al. (2007b)), rather than [valid LCPH(s), valid LCPH(s)+1], but still used a threshold of 0.5 ºC/km. (Please see the Line 223 in the revised manuscript.) Therefore, the DT occurrence frequency based on the new constraint is reduced compared to the original manuscript.

We re-identified the LRT heights using the LRT code provided by Tinney et al. (2022), but still restricted the search range, and the results compared with the bi-Gaussian method are shown in Table 1 (the percentages represent the proportion of temperature profiles in each case), as below. Compared with the results in the original manuscript, both methods work better. The missed detection of LRT (means that there is no value satisfying the definitions within the search range) is reduced, because we calculated the average temperature lapse rate in the revised manuscript by the LRT code provided by Tinney et al. (2022).

**Table 1: Identification results of the bi-Gaussian and LRT. The percentages represent the proportion of temperature profiles in each case. "Missed detection" means that there is no value satisfying the definitions within the search range.**

| Identification results | | Bi-Gaussian | | |
| --- | --- | --- | --- | --- |
| | | Missed detect | ST | DT |
| LRT | Missed detect | 85 (0.11 %) | 174 (0.22 %) | 67 (0.09 %) |
| | ST | 758 (0.96 %) | 54,935 (69.75 %) | **8,682 (11.02 %)** |
| | DT | 257 (0.32 %) | **4,439 (5.64 %)** | 9,362 (11.89 %) |

3) the reasons for choosing 0.5 °C/km as the threshold, as below.

[Figure]

**Figure 1: Temperature profiles from radiosondes in 2014 at Kuqa site (119.7 ºE, 49.25 ºN).**

The evolution of the temperature field at the Kuqa site (119.7 ºE, 49.25 ºN) in 2014 is shown in Fig.1. There are obviously two local temperature minimum layers from January to May along the vertical direction, and the DT structure occurs frequently. With the increase of the surface temperature, the lower local temperature minimum layer was elevated from June to September, prevailing ST structure. After October, it re-evolved into two local temperature minima layers, prompting the formation of DT structures. The DT structures identified by Bi-Gaussian are mainly concentrated in winter and spring, while LRT identified a large proportion of DT structures from May to September. Therefore, we believe that the identification results of bi-Gaussian are more reasonable and more consistent with the evolution process of atmospheric thermal stratifications. In addition, the upper local temperature minima in November and December are too weak to be detected by LRT.

Bi-Gaussian function has a good ability to express the temperature profiles in the UTLS, and is able to more stably and obviously identify the spatial and temporal distribution characteristics of the thermal tropopauses. If a higher threshold is set, some DT structures are difficult to identify, as well as the LRT. As shown in the Fig.1, the threshold of 0.5 ℃/km ensures that bi-Gaussian has a good ability to identify the weak inversion layers. Fig.1 also reflects the differences in the recognition results of the two methods (as listed in Table 4 in the revised manuscript). The LRT identified more DT structures in summer than bi-Gaussian, while the opposite was true in winter.

4) For the 8,682 profiles, defined as DT by bi-Gaussian but only ST by LRT, we made a detailed analysis.

[Figure]

**Figure 2:The cases that were identified as DT by the bi-Gaussian method but only ST by LRT. (a) ratio (contradictory results/ all observation of this site) distribution for the 8,682 temperature profiles. (b) − (h) annual average temperature profiles in 5º latitude bands. A tropopause-based average is adopted by using (DTH1+ DTH2)/2 as a reference level, and −0.5 and 0.5 accurately indicate the locations of DTH1 and DTH2, respectively. (I) the comparison of DTH1 by bi-Gaussian and LRTH for the 8,682 profiles.**

It can be clearly noticed that there is an inversion layer at both DTH1 and DTH2, and the strength of the inversion layer at DTH2 is significantly weaker than that at DTH1, and the bimodality is more pronounced at middle and high latitudes. LRT often fails to capture the weak stability transition (Tinney et al., 2022). The temperature lapse

rate between DTH1 and DTH2 is mainly distributed in [−1.26 ºC/km, −2.54 ºC/km], not satisfying 'the average lapse rate between any level and all higher levels within 1 km exceeds 3 °C/km'. Therefore, the LRT definition of the second tropopause is not satisfied and defines these cases as ST. As can be seen from Fig.2 (I), LRTH is significantly consistent with DTH1, indicating that LRT could accurately identify the first tropopause, even though fail to identify the second tropopause.

2. Figure 8a: I suggest also showing ratio (contradictory results/ all observation of this station) in addition to population.

**Reply:**

Thanks for your suggestion. We have revised the expression.

3. line 287: 'peak at 0.5… peak at -0.5': I find it difficult to understand this normalized height. Does this mean that for all first DT, the altitude is -0.5, and for all second DT, the altitude is 0.5? Could you please add more explanation to the text? Or maybe do not define '0.5' and '-0.5', just use text 'first DT' and 'second DT', because 0.5 maybe misleading, looks like 0.5 km.

**Reply:**

Thanks for your suggestion. We have revised the expression. A description of the normalization process has also been added to the figure caption.

4. Line 288 'the bimodality become gradually unobvious poleward': in figure 8a, it looks like that the DT events happens more frequently at higher latitude, but the bimodality is not very clear? Could this be a result of overestimating DT events over higher latitude? Could you please add more discussion on this result?

1) We greatly admire your keen insight. We did make an error in calculating the average temperature profiles with tropopause-based height, missing some temperature profiles satisfying the conditions, but it is not related to the results identified by the bi-Gaussian method.

2) We deleted 'the bimodality become gradually unobvious poleward' to avoid ambiguity.

3) More discussions can be found in response to Comment 1, which have been added to the revised manuscript.

5. In addition to the tropopause definition described in this work, in the subtropics, dynamics tropopause based on PV or PV gradient is also widely used to detect the transport between stratosphere and troposphere. This paper also describes PV field in section 5, and shows a good agreement between the PV field and the tropopause. It will be interesting to add discussion regarding the dynamics tropopause.

**Reply:**

Potential vorticity (PV) is commonly used to define a dynamic tropopause definition, and a 2 potential vorticity unit (PVU; $1PVU=10^{-6}$ $m^2$ K $kg^{-1}$ $s^{-1}$) value is often chosen to represent the tropopause in synoptic-scale studies (Gettelman et al., 2011; Kunz et al., 2011). PV considers both the atmospheric static stability and the three-dimensional horizontal motion, ensuring the continuity of the dynamical tropopause distribution over large temporal and spatial scales. Therefore, PV is a well–characterized tracer in the study of stratosphere–troposphere interactions (Kunz et al., 2011).

As shown in Fig.14 in the revised manuscript, the 2 PV-based dynamic tropopause is significantly consistent with the thermal tropopause based on bi-Gaussian, which also proves the feasibility and accuracy of the bi-Gaussian method.

Therefore, for the discussion in Sec.5 of this manuscript, we use the PV as an important reference for studying the atmospheric motion of the middle and upper level, which will help us to deeply understand the formation mechanism of the double tropopause structures. Relevant discussions have added to the Discussion in Sec. 5.

**Minor comments:**

1. Figures 1, 9, 10, 11: please add longitude and latitude to your map

**Reply:**

We have added the longitude and latitude labels in all figures if needed, as shown below.

[Figure]

**Figure 3: Spatial distribution of sounding sites (dots) and relevant terrain heights (color-coded).**

2. Figures 6, 8, 9, 10: text is blur

**Reply:**

We have replaced these figures with blur text.

3. Line 9-11: I would say 'the tropopause is an important transition layer, and can be a diagnostic of ..'

**Reply:**

Thanks for your suggestion. We have revised the expression.

4. Line 23-24: 'in the latitude range [16°N, 50°N] ' and 'at mid-latitudes [30°N, 40°N]': please use the same format. This problem in also in lines 300-308.

**Reply:**

Thanks for your suggestion. We have revised the expression throughout the whole manuscript.

5. Line 41: ', climate model simulations': replace ',' into 'and'

**Reply:**

Thanks for your suggestion. We have revised the expression.

6. Line 58: 'buoyancy frequency N has been introduced', not 'introduced' because it is already defined earlier. It is just 'used'.

**Reply:**

Thanks for your suggestion. We have revised the expression.

7. Line 66: 'from different monsoon circulation systems, such as the Asian summer monsoon and polar vortex..', polar vortex is not a monsoon circulation system

**Reply:**

Thanks for your suggestion. We have modified 'different monsoon circulation systems' to 'atmospheric circulation systems'.

8. Line 101: 'as described in detail in the literature (Guo et al., 2016)'

**Reply:**

We have modified 'as described in detail in the literature (Guo et al., 2016)' to 'as described in Guo et al. (2016)'.

9. Line 102: 'representing an excellent opportunity', sounds wired, consider use 'providing an excellent opportunity.'

**Reply:**

Thanks for your suggestion. We have revised the expression.

10. Table 1: incorporate the most important information in the text. For example, mention that the vertical resolution is 5-8 m when saying 'higher vertical resolution than reanalysis'.

**Reply:**

Thanks for your suggestion. We have revised the expression. 'Once- or twice-daily radiosondes, launched at 08:00 and 20:00 (local time), throughout four seasons have a higher vertical resolution (about 5 to 8 m) than reanalyses.'

11. Line 110: 'there are 37 vertical layers from 1000 hPa to 1 hPa': what is the vertical resolution over the levels of interest in this paper?

**Reply:**

We have modified 'there are 37 vertical layers from 1000 hPa to 1 hPa' to 'there are 23 layers in vertical direction, including 850, 825, 800, 775, 750, 700, 650, 600, 550, 500, 450, 400, 350, 300, 250, 225, 200, 175, 150, 125, 100, 70, 50 hPa.'.

12. Line 119: between the upper (stratosphere) and lower (troposphere) parts: this is ambiguous, consider just say between the stratosphere and troposphere, or stratosphere (upper altitude) and troposphere (lower altitude).
between the stratosphere and troposphere

**Reply:**

We have modified 'between the upper (stratosphere) and lower (troposphere)' to 'between the stratosphere and troposphere'.

13. Line 125: 'in view of' this sentence is blur. The three sentences in this paragraph are: 'in view of the fact.. in addition' are both a description previous works, but 'next' follows by the work in this paper. This logic flow does not make sense.

**Reply:**

Thanks for your suggestion. We have revised the expression. The five thermodynamic definitions are first described and the deviation of the identification results for the same profiles is shown to elicit the necessity for the proposed new method.

14. Section 2.3.1 the title is 'tropical tropopause layer', but in figure 2 it is a subtropical station

**Reply:**

Thanks for your suggestion. We have modified the content of Section 2.3, by replacing '2.3.1 Tropical tropopause layer' to '2.3.1 Cold point tropopause and potential temperature lapse rate minimum tropopause', replacing '2.3.2 Extratropical tropopause' to '2.3.2 Lapse rate tropopause', adding '2.3.3 Curve fitting to Brunt-Vaisala frequency' and '2.3.4 Potential temperature gradient tropopause'.

In the revised manuscript, we replaced the data from a tropical site (112.33 °E, 16.83 °N), and the five methods were applied to the temperature profiles. The results are shown in Fig. 4 as below.

[Figure]

Figure 4: Five different thermal definitions were shown in a tropical site (112.33 °E, 16.83 °N).

15. Line 326: 'Bi-Gaussian method prefers to define higher and colder temperature inversion layer as DTH2, which leads to an increase in the occurrence frequency and thickness of DT': could you explain more about this logic chain?

**Reply:**

Similar to that expressed in Figure 8. However, the expression in the original manuscript does lack clarity and has been deleted.

16. Line 344: 'tropopause height has an increasing trend under global warming' I doubt this value may not be very large, could you please quantify it according to the estimation in your reference?

**Reply:**

As reported in Meng et al. (2021), tropopause heights increase about 50 to 60 m/decade, which really isn't a great value.

17. Line 345: 'the change trend' 'trend' is misleading with the trend in the last paragraph. It sounds like a trend changing with time.

**Reply:**

We have modified 'The change trend of tropopause height from tropical to subtropical regions is discontinuous' to 'The meridional distribution of tropopause height from tropical to subtropical regions is discontinuous'

18. Line 365: '(Buchart, 2022)' this citation is not included in the reference. I didn't check the full reference list and the authors should double check it to make sure this mistake does not happen again.

**Reply:**

Thanks for your careful check. We have checked all references to make sure this mistake does not happen again.

19. Section 5: I suggest use another name instead of 'discussion', since this is an important part of this paper, not a discussion.

**Reply:**

Thank you for your suggestion. We have modified the caption title to 'Discussion on the formation mechanism of double tropopause in mid-latitude region in winter'.

**Table 1: Identification results of the bi-Gaussian and LRT. The percentages represent the proportion of temperature profiles in each case. "Missed detection" means that there is no value satisfying the definitions within the search range.**

| Identification results | | Bi-Gaussian | | |
|---|---|---|---|---|
| | | Missed detect | ST | DT |
| LRT | Missed detect | 85 (0.11 %) | 174 (0.22 %) | 67 (0.09 %) |
| | ST | 758 (0.96 %) | 54,935 (69.75 %) | **8,682 (11.02 %)** |
| | DT | 257 (0.32 %) | **4,439 (5.64 %)** | 9,362 (11.89 %) |

According to the latest statistics, bi-Gaussian method and LRT fails to identify a tropopause in 1.39 % and 0.42 % of profiles, slightly higher than the missed detection rate of LRT. There is no failure rate of about 12.5% that you mentioned, even in the original manuscript.

4) In the revised manuscript, it is clarified that the LRT algorithm is also limited by the search range, and the meaning of 'Missed detection' is labelled in Table 4.

2. Second, the physical meaning and justification of a definition based on local minima in temperature is not clear and otherwise presumed to be weak. Past literature demonstrates thoroughly that cold-point definitions are not appropriate outside of the tropics and even in the tropics commonly result in identification of a level that is not dynamically or chemically relevant to the purposed use of such a definition – to accurately identify the bound (or transition zone) between troposphere and stratosphere air. Conversely, temperature minima near and above often result from convection and wave activity and can be an important failure mode for some existing definitions. To rely upon an error-prone basis of tropopause definition as local temperature minima provide is therefore highly questionable. Moreover, because the LRT has been comprehensively demonstrated to be reliable most of the time, differences between the LRT and any proposed definition solicit increased scrutiny of an alternative definition. It must be clearly demonstrated why an alternative definition is more reliable than the

LRT or a similarly reliable definition (there are recent relevant studies not cited), otherwise the exercise presents simply a difference without explanation or significance.

**Reply:**

1) The CPT is defined by the minimum in the temperature profile and marks a sharp increase in stability, above which the potential temperature profile is close to radiative equilibrium (Gettelman and Forster, 2002; Randel and Park, 2019; Pan et al., 2018; Pan et al., 2014). CPT definition has better applicability in the tropics because of the simpler vertical structure of atmospheric temperatures in the tropics, with fewer atmospheric temperature inversions. In other words, CPT is still highly reliable for identifying single tropopause, and its limitations will be exposed in multiple tropopause structures. By definition of the CPT, CPT can only return one identification result for both single and multiple structures, which is exactly why CPT cannot identify multiple structures. Therefore, we use local cold point instead of CPT, and only the local cold points that have passed the significance test are considered to be the tropopause heights.

2) In order to avoid ambiguity, "local coldest point (LCP)" was replaced to "possible tropopause height (PTH)" in the revised manuscript. There are relatively strong peaks at PTHs, and these PTHs actually contain the height layers that satisfy the LRT definition. Firstly, find all the PTHs in the search range, which can reduce the missed detect rate, and then the fitting optimal solution is obtained by bi-Gaussian function fitting. The recognition process is similar to LRT. And, the bi-Gaussian function has a good ability to express the temperature profiles in the UTLS. Therefore, bi-Gaussian is not a substitute for CPT, but has a more reasonable identification result than CPT, and can capture a more complete evolution process.

3) We made a detailed comparative analysis according to the identification of bi-Gaussian and LRT, including the same and contradictory results.

(i) There is remarkable consistency between bi-Gaussian and LRT for the cases of same identification results, but bi-Gaussian is slightly larger than the LRT, which may be determined by the inherent properties of the two definitions (see Fig.S5 in the Supplement).

(ii) Especially in view of the contradictory cases, some examples and statistical analysis are shown to explain for the contradiction. Please refer to Fig.S1–Fig.S4 in the Supplement and the Fig.8 in the main text.

4) Compared with CPT, bi-Gaussian improves accuracy. Specifically, bi-Gaussian can not only identify the double tropopauses, but also identify the same temperature inversion layer as LRT, and the identification results have less deviation from LRT, as shown in Fig.1 as below. The bias between CPT and LRT is distributed at [0.31 km, 1.84 km], while the bias between bi-Gaussian and LRT is stable at 0.37 km, even at mid-latitudes.

[Figure]

**Figure 1: The biases of bi-Gaussian and CPT against LRT in different latitudes.**

3. Third, unless a definition is created to serve a very specific region or purpose, I consider global comparisons of a new tropopause definition with existing ones to be a necessary element of such a study. The narrow focus on China in this study is thus a major shortcoming given the aim of the effort.

**Reply:**

1) The radiosonde data in China used in the manuscript cover tropical, subtropical, temperate and plateau climate zones, which can verify the feasibility and accuracy of the bi-Gaussian method.

2) In the next research work, we plan to further analyze the global tropopause vertical structure with more extensive data covering a wider area, such as GPS occultation.

4. Fourth, while I do sincerely appreciate the authors' attempt to identify multiple tropopauses since only two proven definitions currently do so, the result that double tropopause occurrences increase from ~3% based on the LRT to more than 70% with the new definition is extremely concerning. Namely, as is true and necessary for any tropopause definition, an identified tropopause (primary or otherwise) in a profile should have an important physical or dynamical meaning. Otherwise, you attain nothing but vast identification of arbitrary levels that happen to have a local minimum in temperature. I do not expect the authors to demonstrate physical or dynamical linkages for their multiple tropopause definitions, but such have been well documented for double tropopauses that result from the LRT definition. The fact that you see such a tremendous increase casts serious doubt on the potential utility of such a definition, especially because it has also been demonstrated that multiple tropopauses identified by the LRT do not always have a clear physical or dynamical explanation.

**Reply:**

1) According to the statistical results listed in Table 4, the DT occurrence frequency defined by LRT and bi-Gaussian is 16.89% and 27.69 %, respectively.

2) In order to avoid misjudgment of the tropopause structure due to local temperature fluctuations caused by atmospheric fluctuations, we have improved constraints in the significance test, by changing the range of the linear fitting to [valid LCPH(s), valid LCPH(s)+2] (referring to Randel et al. (2007b)), rather than [valid LCPH(s), valid LCPH(s)+1], but still used a threshold of 0.5 ℃/km. Therefore, the occurrence frequency of DT in the revised manuscript decreased. Spatial distribution characteristics of DT occurrence frequency in China based on bi-Gaussian method are shown in the Fig.11 in the revised manuscript. The maximum of annual mean occurrence frequency (thickness) is about 47.19 % (5.42 km), and the minimum is about 1.07 % (1.96 km) in the latitude range of [16 ºN, 50 ºN]. And, DT occurs most frequently in mid-latitude regions in winter. The meridian distribution of the tropopause based on the bi-Gaussian method is qualitatively and quantitatively consistent with the previously reported results (Randel et al., 2007a; Peevey et al., 2012; Xu et al., 2014).

5. Fifth, an important – though not mandatory – expectation for a tropopause definition is that its application is straightforward and not prone to confusion or misuse by others. The proposed definition is quite complicated, with many conditional steps that are likely to be inconsistently and inappropriately applied by others. Thus, simplification of the procedure should be a priority. Moreover, it is never specified what units are used for the conditional elements of the proposed definition, which are ultimately necessary for others to replicate this work in the future.

**Reply:**

According to the latest identification results of bi-Gaussian, it has a good agreement with LRT, and can stably identify the continuous evolution process of the thermal tropopause structures. In the following research, we will optimize and simplify the algorithm, such as realizing the identification of triple tropopauses.

**Specific Comments**

1. Because of the substantial concerns I have with the design and execution of the study, I will not list myriad technical corrections here, but highlight some additional problematic statements or impressions. There are multiple recent efforts to develop tropopause definitions that are not acknowledged or cited. There are also many other contextual works that would help greatly in the presentation, framing, justification, and discussion of such work. I encourage the authors to dive deeper into literature review to improve upon these issues, which will help direct future efforts towards accomplishing this study or another iteration.

**Reply:**

We sincerely appreciate the valuable comment. As suggested by the reviewer, we have checked the literature carefully and added more references (e.g. Pan et al. (2004), Maddox and Mullendore (2018), Tinney et al. (2022), Boothe and Homeyer (2017), and Ivanova (2013)) about developing the tropopause definitions into INTRODUCTION part in the revised manuscript.

2. Lines 34-36: the tropopause does not perform a role in stratosphere-troposphere exchange (STE), but its definition is required to assess it; dynamic mechanisms are the role for STE.

**Reply:**

Thanks for your suggestion. We have rewritten it to 'The tropopause is a transitional layer between the upper troposphere-lower stratosphere (UTLS)'.

3. Line 39: there are many other (and increasingly comprehensive) studies of the tropopause and its relation to climate change that are not cited.

**Reply:**

We sincerely appreciate the valuable comment. We have checked the literatures carefully and added more references (e.g. Meng et al. (2021), Xian and Fu (2017), Seidel et al. (2001), Shepherd (2002), Seidel and Randel (2006), and Thompson et al. (2021)) on the relationship between tropopause and climate change into INTRODUCTION part in the revised manuscript.

4. Line 60: should acknowledge here and elsewhere that this "cliff-like decline" is broadly recognized as the "tropopause break" and cite additional work.

**Reply:**

We have modified "cliff-like decline" to "tropopause break", and cited relevant literatures (e.g. Xian and Homeyer (2019), Rieckh et al. (2014), Palmen (1948), and Randel et al. (2007a)).

5. Line 77: "subject to controversy" is overstated. It would be better described as "active areas of research"

**Reply:**

Thanks for your comment. We have revised it as the comment.

6. Lines 94-96: this is not appropriate motivation for the use of radiosonde observations. Radiosonde observations are the traditional and most widely used data for studying the upper troposphere and lower stratosphere and defining the tropopause.

**Reply:**

  Thanks for your suggestion. We have rewritten the sentence as "Radiosonde observations of air temperature, the traditional and most widely used, are crucial and essential for studying the fine tropopause structures".

7. Line 102: this is also a very unusual introductory statement and motivation. There have been multiple well-cited studies that demonstrate why double tropopauses are frequent in the midlatitudes.

**Reply:**

  We have deleted "In order to investigate why DT structures in the mid-latitudes are frequent,".

8. Lines 106-117: this is presented suddenly and without explanation of its significance and intended use.

**Reply:**

  We have re-written this part according to your suggestion. Transitional sentence, "The stratospheric polar vortex is a large-scale circulation system over the polar region in the Northern Hemisphere winter, which is related to tropospheric circulation anomalies and plays an important role in the stratosphere–troposphere coupling (Ren and Cai, 2007; Zhang et al., 2016; Liang et al., 2023)." has been added at the beginning of the paragraph to introduce "the polar vortex intensity" and "ERA5 reanalysis".

9. Line 119: this statement is not true in multiple ways and is contradicted throughout the article. Most existing tropopause definitions have been demonstrated to be chemically, physically and/or dynamically meaningful. At least two existing definitions have been demonstrated to be universal – the LRT and the recently-developed potential temperature gradient tropopause (PTGT) definition.

**Reply:**

Thanks for your suggestion. We have rewritten this part. And, the potential temperature gradient tropopause (PTGT) definition was cited in the revised manuscript.

10. Figure 2: are the lines in panels (b)-(d) averages? This analysis is not well explained or described.

**Reply:**

The red lines in panels (b)-(d) do indicate the mean profiles. More information for analysis was described.

11. Section 2.3.2: there are several issues here. First, it is presented as though only the Brunt-Väisälä frequency is used for tropopause definition. Second, one curve-fitting method from a single study (Homeyer et al. 2010) is used without explanation that such is the source.

**Reply:**

1) We have modified the content of Section 2.3, by replacing '2.3.1 Tropical tropopause layer' to '2.3.1 Cold point tropopause and potential temperature lapse rate minimum tropopause', replacing '2.3.2 Extratropical tropopause' to '2.3.2 Lapse rate tropopause', adding '2.3.3 Curve fitting to Brunt-Vaisala frequency' and '2.3.4 Potential temperature gradient tropopause'.

2) We have emphasized in the revised manuscript that curve-fitting method is proposed by Homeyer et al. (2010).

12. Line 183: TH is not defined and is difficult to follow its meaning here and after.

**Reply:**

We have replaced 'TH' with 'tropopause height' and deleted all 'TH' abbreviations in the text.

13. Line 225: extremely overstated. A high correlation for the fitting process does not demonstrate potential for accurate tropopause definition, but rather that you have success at identifying local temperature minima.

**Reply:**

1) We only want to use R2 to evaluate the expression ability of the bi-Gaussian function to atmospheric temperature profiles in UTLS. Higher R2 indicate better goodness of the bi-Gaussian function. R2 is greater than 0.8 in at least 90% temperature profiles, and the average R2 of all profiles reaches 0.9. Consequently, the bi-Gaussian function exhibits remarkable potential for accurately explicating temperature profiles in UTLS, ensuring that LCPs are successfully identified.

2) In the revised manuscript, we have deleted the exaggerated description of R2 in the comparison of LRT and bi-Gaussian.

**Reference**

[revised manuscript text omitted]

**RC3:**

**General Comments:**

This paper is an interesting study and methodology of the definition and computation of single and multiple tropopauses (TPs). Although the authors have made intensive investigation with the excellent radiosonde measuring net in China, however, the results of the bi-Gaussian fitting method (BGF) are not convincing for me and with respect to the publication requirements of ACP.

I have listed below various items regarding the analysis and especially the form of the presentation of the study, which needs further improvements. If the authors consider most of the comments in a revised version, the article may be acceptable for publication in ACP.

In general, I have concerns about the quality of the BGF method. The study misses a validation of the tropopause results with respect to frequency (double TP events) and especially height of the tropopause (TPH) with independent measurements (e.g. GPS occultation) and methods. Although, this is partly done in Fig. 6, I was a bit puzzled that later, differences of >1km between the tropical TPH of lapse rate TP and BGF are described with 'small'. It is already obvious from former studies that the LRT is usually placed below the cold point in the tropics. So, why do you compare apples and oranges? Consequently, I was a bit surprised that Fig.6 shows no clear indication for a positive bias (are most of the profiles not really in the tropics?), but many TPHs are quite high (>17 km), which looks very tropics-like. However, a closer look seems to show such a 'positive' bias in Fig. 6 for STH/DTH1 compared to LRTH1. This fact is not discussed properly in the manuscript with respect to different definitions of both TP methods.

**Reply:**

1) Spatial distribution characteristics of DT occurrence frequency in China based on bi-Gaussian method are shown in the Fig.9 in the revised manuscript. The maximum of annual mean occurrence frequency (thickness) is about 47.19 % (5.42 km), and the minimum is about 1.07 % (1.96 km) in the latitude range of [16 ºN, 50 ºN]. And, DT

occurs most frequently in mid-latitude regions in winter. The meridian distribution of the tropopause based on the bi-Gaussian method is qualitatively and quantitatively consistent with the previously reported results (Randel et al., 2007a; Peevey et al., 2012; Xu et al., 2014), including a research based on GPS radio occultation.

2) In order to avoid ambiguity, "local coldest point (LCP)" was replaced to "possible tropopause height (PTH)" in the revised manuscript. There are relatively strong peaks at PTHs, and these PTHs actually contain the height layers that satisfy the LRT definition. Firstly, find all the PTHs in the search range, which can reduce the missed detect rate, and then the fitting optimal solution is obtained by bi-Gaussian function fitting. The recognition process is similar to LRT. And, the bi-Gaussian function has a good ability to express the temperature profiles in the UTLS. Therefore, bi-Gaussian is not a substitute for CPT, but has a more reasonable identification result than CPT, and can capture a more complete evolution process.

3) The CPT altitude is on average 400 m higher than the LRT with values varying between 300 m in July and 500 m in September (Schmidt et al., 2004), the CPT and LRT height definitions are inconsistent, with a difference of 2 km considered to be the boundary value (Pan et al., 2018; Xia et al., 2021). On the one hand, this difference is caused by the inherent properties of the two definitions (see the Fig. 1 below), because CPT is the transition point at which temperature lapse rate changes from negative to positive, which is common in the tropics. On the other hand, CPT defines the higher and colder inversion layer (if exist) as the tropopause, so that the two methods can't identify the same temperature inversion layer (see Fig. 3(a) in the revised manuscript). This situation mostly occurs in the middle and high latitudes, which may be one of the reasons for the large deviation between CPT and LRT in the middle and high latitudes.

[Figure]

**Figure 1: An example to explain the inherent bias of CPT height over LRT height.**

4) The CPT is defined by the minimum in the temperature profile and marks a sharp increase in stability, above which the potential temperature profile is close to radiative equilibrium (Gettelman and Forster, 2002; Randel and Park, 2019; Pan et al., 2018; Pan et al., 2014). CPT definition has better applicability in the tropics because of the simpler vertical structure of atmospheric temperatures in the tropics, with fewer atmospheric temperature inversions. In other words, CPT is still highly reliable for identifying single tropopause, and its limitations will be exposed in multiple tropopause structures. According to CPT definition, CPT can only return one identification result for both single and multiple structures, which is exactly why CPT cannot identify multiple structures. Therefore, we define the local coldest point(s) instead of CPT in

the new bi-Gaussian method as the possible tropopause height(s), and only the local coldest point(s) that have passed the significance test are considered to be the tropopause heights.

5) Compared with CPT, bi-Gaussian improves accuracy. Specifically, bi-Gaussian can not only identify the double tropopauses, but also identify the same temperature inversion layer as LRT, and the identification results have less deviation from LRT, as shown in Fig.2 as below. The bias between CPT and LRT is distributed at [0.31 km, 1.84 km], while the bias between bi-Gaussian and LRT is stable at 0.37 km, even at mid-latitudes.

[Figure]

**Figure 2: The biases of bi-Gaussian and CPT against LRT in different latitudes.**

6) In some mid-latitude areas, the single tropopause height can be elevated 17km in summer, such as the Tibetan Plateau. So, many TPHs are quite high (>17 km), which looks very tropics-like.

**Detailed Comments:**

1. L11: 'physiochemical' unusual wording, please change.

**Reply:**

Thanks for your suggestion. We have modified 'physiochemical coupling' to 'physical–chemical coupling'.

2. L15: 'in mathematical statistics' not clear to me why this term is necessary.

**Reply:**

We have deleted 'in mathematical statistics'.

3. L37: 'stratosphere vis this "gate"' Is vis really the correct wording here?

**Reply:**

Thanks for your suggestion. We have modified 'vis' to 'through'.

4. L40: delete 'in'

**Reply:**

Thanks for your suggestion. We have revised it according to the comment.

5. L46: 'concept of the dynamical tropopause'

**Reply:**

Thanks for your suggestion. We have revised it according to the comment.

6. L55: lapse rate minimum tropopause (LRM)

**Reply:**

Thanks for your suggestion. We have revised it according to the comment.

7. L57: gauge -> estimate

**Reply:**

Thanks for your suggestion. We have revised it according to the comment.

8. L59: what do you mean with 'ideal models'? please clarify.

**Reply:**

Here, we refer to the sentence 'Idealized models of the tropopause usually assume a discrete jump in the static stability from relatively low values in the well-mixed troposphere to high values in the stable stratosphere' in Homeyer et al. (2010). To avoid misunderstandings, we have deleted the 'ideal models'.

9. L74: 'key stratification' sounds misleading to me.

**Reply:**

We have modified 'key stratification' to 'key transition layer'.

10. L128: Is there a lower boundary of the tropopause? Please, clarify.

**Reply:**

According to the results in Gettelman and Forster (2002), the upper and lower boundary of the tropical tropopause layer is well characterized by the CPT and LRM, respectively. We rewritten the sentence: 'In addition, the CPT and LRM are also adopted to characterize the upper and lower boundaries of the tropical tropopause layer.'

11. L129: 'four' I count only three TP definitions (LRT, CPT, and N^2). Please, clarify.

**Reply:**

In the previous manuscript, four definitions were CPT, LRM, LRT, and N2 (as shown in Fig. 2 in the revised manuscript). In the revised manuscript, we added the PTGT method, so there are five methods in total in Fig. 2.

12. L143: delete 'And': The cold point ...

**Reply:**

Thanks for your suggestion. We have revised it according to the comment.

13. L166: what is different? Please be more specific with your statements.

**Reply:**

We have rewritten the sentence as 'the tropopause heights identified by the above five definitions are quite different'.

14. L167: please correct, 'close to the CPTH'

**Reply:**

Thanks for your suggestion. We have revised it according to the comment.

15. L171: 'highly effective' for what? Do you mean the methods?

**Reply:**

We have rewritten it as 'CPT and LRT have good applicability in the tropics'.

16. L172: … in the extratropics …

**Reply:**

Thanks for your suggestion. We have revised it according to the comment.

17. L176: DT, you may have to introduce DT not only in the Abstract but also in main text.

**Reply:**

Thanks for your suggestion. We have revised it according to the comment.

18. L200-215: How do you handle triple structures of the TP? Is the method robust, does it detect the upper or lower 2nd TP?

**Reply:**

What cannot be ignored is the presence of triple tropopauses, even if the occurrence frequency of triple tropopauses is very low. The third tropopause is mainly distributed at ~50 hPa (Anel et al., 2007; Xu et al., 2014), so we assume that there are double tropopauses at most in the search range. This is one of the important reasons for constraining the search range. An example can be referred to in Fig. S3 in the Supplement.

19. L220: The parameter of the formula of Table 3 are frequently used in the manuscript. Consequently, they must be introduced in text and not in the table, as well as a more detailed description is necessary.

**Reply:**

Thanks for your suggestion. We have revised it according to the comment.

20. L230: I cannot follow the arguments on R^2 and why this number should give me confidence that the TP is detected correctly. It's just the quality of the fit. It is necessary to check the quality of the fit.

**Reply:**

1) We only want to use $R^2$ to evaluate the expression ability of the bi-Gaussian function to atmospheric temperature profiles in UTLS. Higher $R^2$ indicate better goodness of the bi-Gaussian function. $R^2$ is greater than 0.8 in at least 90% temperature profiles, and the average $R^2$ of all profiles reaches 0.9. Consequently, the bi-Gaussian function exhibits remarkable potential for accurately explicating temperature profiles in UTLS, ensuring that LCPs are successfully identified.

2) In the revised manuscript, we have deleted the exaggerated description of $R^2$ in the comparison of LRT and bi-Gaussian.

21. L254: 'darkest patches' ? Red is not dark compared to blue. 'The majority of the events are located on the …'

**Reply:**

Thanks for your suggestion. We have revised it according to the comment. 'The majority of the distribution are located on the line y=x.'

22. L274: Please reword the sentence. It is not clear to me what you like to say. Why is a threshold critical for an accurate result of the TP? It's part of the definition.

**Reply:**

According to the LRT definition, (–2 ℃/km, 2 km) and (–3 ℃/km, 1km) are the lapse rate and thickness thresholds for the first and second tropopause, respectively. Sensitivity test of lapse rate and thickness thresholds to tropopause estimates on LRT criteria was performed in Hoffmann and Spang (2022). The statistical analysis of the lapse rates from the middle troposphere to the lower stratosphere suggests that the thermal tropopause critically depends on the lapse rate threshold and the layer depth applied in the WMO definition.

Of course, the lapse rate and thickness thresholds ((–2 ºC/km, 2 km) and (–3 ºC/km, 1km)) are not absolutely universal, because lapse rates at specific locations may indicate different levels of stability. Since the strength of stability plays an important role in convective transport, it is worthwhile to note that a fixed temperature lapse rate does not necessarily correspond to a fixed stability threshold (or vice versa) (Maddox and Mullendore, 2018).

We have deleted the sentence in the revised manuscript.

23. L282: by the bi-Gaussian method, but only ST by LRT.

**Reply:**

Thanks for your suggestion. We have revised it according to the comment.

24. L280: I cannot really follow the description and conclusions of Fig. 8. I would suggest writing the whole section and caption new. More details on the methods are necessary. Why are both TPHs constant on +/- 0.5 units? The normalization is not really described in detail and difficult to follow. The arguments with $R^2$ are again very confusing.

**Reply:**

25. L300: new section and subsection, please introduce 'the occurrence frequency' of what kind of parameter?

**Reply:**

We have modified to 'Double tropopauses structures: occurrence frequency and thickness'.

26. L307: Please rewrite this sentence 'The thickness …'. I can't get a handle on the terms 'latitudinal plain' and 'giant topography'.

**Reply:**

We have modified to 'The thickness in the area [90 °E−100 °E, 26 °N−32 °N] is obviously greater than that of the adjacent plain in same latitude, which may be resulted from the complex topography of the Tibetan Plateau'.

27. L333-344: The discussion is misleading. It is always clear that CPT and LRT will not deliver the same tropopause height due to the definition of both parameters. In the tropics there should be an offset, and this becomes obvious in your Fig. 6. Of course you can show these comparisons, but it is no proof about your TP determination, because the comparison works with 'apple and oranges'. You may quit this part.

**Reply:**

As discussed above, bi-Gaussian is not a replacement for CPT, but an upgraded and improved identification method. Therefore, we have kept some statements and removed some unnecessary comparisons.

28. L352: Not the TP is a source of gravity waves but processes in the TP region trigger GW formation.

**Reply:**

We have revised to "Tibetan Plateau, a source of gravity waves (Hoffmann et al., 2013; Khan et al., 2016), may be one of the contributors to the asymmetry between the northern and southern hemispheres".

29. L364: Here went something wrong 'atmospheric dynamic processes …', please reword.

**Reply:**

Thanks for your correction. The sentence has been revised.

30. L382: '… and high static stability of the air masses creates …'

**Reply:**

Thanks for your correction. The sentence has been revised.

31. L427: delete 'which is more than half …'

**Reply:**

Thanks for your correction. The sentence has been deleted.

32. L432: Is TT1 introduced before?

**Reply:**

"TT1" was replaced to "DTT1" in the revised manuscript.

33. L435: and increases DTT2.

**Reply:**

Thanks for your correction. The word has been revised.

34. L437: '… intensifies the atmospheric mixing' may be better.

**Reply:**

Thanks for your correction. The word has been revised.

35. L442: I have doubts that the argumentation with R^2 makes sense, especially in the conclusion section (see above and concerns by other reviewers).

**Reply:**

We re-written the Conclusion, and deleted some confusing sentences.

36. L449: Again, I cannot follow the argument '… ambiguity of LRT constrained by thresholds.'. The bi-Gaussian method is not constrained by thresholds but by the bi-Gaussian fit approach and the quality of the fit, which is also very likely a threshold criterium.

**Reply:**

We re-written the Conclusion, and deleted some ambiguous sentences.

37. L470: formatted -> formed

**Reply:**

Thanks for your correction. The word has been revised.

Technical issues:

1. Most of figures show a lack in resolution, which makes it difficult to read numbers and figure legends. For publication this needs definitely a substantial improvement (Fig 1, 5, 6, 7 -10)

**Reply:**

We have updated all Figures with low resolution.

2. Fig 3: please enlarge the figure and especially the font size. What do mean with 'Modal'? This is not used in the text, please change this term.

**Reply:**

We have updated Fig.3 with your suggestion. The "Modal 1" and "Modal 2" are defined at first appearance, which are two modals for bi-Gaussian function.

3. Fig 3b: What is the red sub-plot in (b), Temp versus Altitude. This is not explained in the Figure capture and makes no sense to me at all. If possible, just delete it.

**Reply:**

We have updated Fig.3 with your suggestion, deleting the sub-plot.

4. Fig. 5: Fonts are far too small!

**Reply:**

We have updated Fig.5 with your suggestion.

5. Fig. 7: please, enlarge the text fonts (e.g. dT/dz). In addition, there seems something wrong in the wording 'Case A indicates that presents …'. Could it be better: 'Case A indicates the presence of a higher …'

**Reply:**

We have updated Fig.7 with your suggestion.

6. Fig.10: For me it would be better to use identical TP height ranges for all three TPHs. The color code is misleading, e.g. why should STH be higher than DTH2, but it is just the color code? Again, the resolution of the figure is not good enough. It is not possible to read all the letters and numbers properly.

**Reply:**

We have updated the Fig.10, using identical ranges. Meanwhile, the resolution was improved.

7. Fig 12a: Is this PV plot presented for a specific theta level? If so, please add this important information.

**Reply:**

Thanks for your careful check. "at 315 K isentropic surface" have added to the figure caption and title of the colorbar in Fig.12(a).

**Reference**

[revised manuscript text omitted]

---

## Referee Report (RR1)

**Review of "A novel method to detect the tropopause structure based on bi-Gaussian function"**

The authors have put a solid amount of work into revising the manuscript to address my comments and those of the other reviewers. The clarity of the paper and its assertions have improved and many of my concerns have been reduced. However, there are still a handful of edits that I believe must be made before this can be considered for publication. Most are technical in nature, but there are two general points remaining that are most important in my view.

**General Comments**

1. The use of "local cold**est** point(s)" throughout is unnecessarily confusing. This would be much clearer stated as "local cold point(s)", since only one could possibly be the coldest.

2. Units for all equations and terms outlining the bi-Gaussian method are still absent from the paper. This includes Equations 5 & 6, where the resulting altitude units and dependent latitude units should be clearly defined, line 220 where the PTH height units for the search range should be defined, everything in Table 2, and everything in Table 3.

3. Section 5 is somewhat vague, unclear, and includes a lot of hand-waving examples of double tropopause characteristics and purported linkages to other metrics. It is neither convincing or necessary based on my evaluation and I recommend simply removing it (lines 403-481; Figs. 13-15; Table 5) and related lines 503-507 from the manuscript.

**Technical Edits**

Line 12 – recommend revising second instance of "structures" to "characteristics.

Line 16 – "tropopauses" should be "tropopause"

Line 18 – "remarkable" should be deleted. This is subjective, non-scientific language.

Line 19 – delete "recognition"

Line 28 – "formatted" should be "formed"

Line 30 – recommend revising "the case of" to "with cases that" & delete "happens occasionally"

Line 37 – "tropopause" should be "tropopauses"

Line 43 – "of tropopause" should be "of the tropopause" & "formatted" should be deleted

Line 48 – "discovered the tropopause" should be "discovered tropopause"

Line 57 – "separated" should be "used"

Line 58 – "lapse minimum rate" should be "lapse rate minimum"

Line 65 – "called" should be "called the"

Line 71 – delete "tracers"

Line 75 – replace "compositions, with active STE" with "constituents"

Line 80 – "tropopause" should be "the tropopause"

Line 85 – "tropopause" should be "the tropopause"

Line 88 – "details" should be "detail" & "with the existing" should be "with existing"

Line 98 – delete "there are"

Line 147 – revise "influenced by stratospheric" to "constrained by"

Line 159 – "gradient" should be "gradients" & "compositions" should be "composition"

Line 164 – "following" should be "follows"

Line 169 – "following" should be "follows"

Line 179 – "LRT, provides" should be "LRT, and provides"

Line 197 – "minimum points" should be "minimum temperature points"

Line 202 – "following" should be "follows"

Line 252 – once again, "remarkable" should be deleted.

Line 261 – "general" should be "total" & the numbers here seem incorrect. If I total the lines from Table 4, I instead get 23.00% and 17.85%.

Line 289 – the dots are "magenta", not "cyan"

Line 309 – "contradictory" should be "contradiction"

Line 315 – insert "with a" before "prevailing ST structure"

Line 320 – start sentence with "The"

Line 322 – "in the Fig." should be "in Fig."

Line 323 – "bi-Gaussian" should be "the bi-Gaussian approach"

Line 335 – "if exist" should be "if multiple exist"

Line 343 – "accuracy" would be better stated as "agreement with the LRT"

Line 400 – "in UTLS" should be "in the UTLS"

Lines 409-411 – I do not know what this sentence means.

Line 427 – what "invades"?

Lines 428-429 – I also do not know what this sentence means.

Line 440 – "at 315 K" should be "at the 315-K"

Lines 444-446 – this sentence does not begin correctly, but I'm not sure I know exactly how you want to relate it to something shown/mentioned previously.

Line 459 – "does" should be "do"

Line 488 – "based bi-Gaussian" should be "based on bi-Gaussian identifications"

Line 491 – delete "monotonous"

Line 500 – delete "and uploading"

---

## Author Response (AR2)

**Author's response**

**Minor revision-egusphere-2024-345-referee-report-1**

**We appreciate the valuable review and constructive feedback provided by the reviewers. We agree with the reviewers' suggestions and carefully revise the manuscript. The detailed corrections are listed below.**

Dear authors,

please consider the following minor revisions for the final version of your manuscript.

1. Figure 2 L130: My impression is that the complete analysis is based on - the only here mentioned - 15-point running mean of (potential) temperature RS profiles. If this is the case, then please note this explicitly in the radiosonde section. This is an important information, even it is not used generally then please explain if you used any other vertical smoothing (what I would expect).

**Reply:**

We have added the sentence "A 15-point running mean was adopted for temperature and potential temperature profiles." in Section 2.1 to introduce the processing of radiosonde data.

2. L163: 'buoyancy frequency' = N (not equal N2).

**Reply:**

Thank you for your comment. We have corrected it.

3. L178: The information on how the PTGT works is too sparse. Please add information here.

**Reply:**

We have added more information about the PTGT method, such as specific definitions, in Section 2.4.

4. L200ff: Can you please explain in more detail why gravity wave with small vertical wavelength (< 2-4 km) shall not produce unrealistic 2nd and potentially 3rd tropopause events with your applied methods

I am also wondering why you introduce an additional TP detection, although you stay finally with LRT for comparison.

**Reply:**

1) Small-scale temperature fluctuations, resulting from small vertical–wavelength gravity wave perturbations (Rechou et al., 2014), may be defined as the local cold point(s). Firstly, some local cold points due to gravity waves with small vertical wavelength will be filtered out during the double Gaussian function fitting process in the whole search range.

Secondly, we set an inversion layer strength (0.5 ºC/km) as a threshold for significance testing, so insignificant LCP(s) will be excluded.

Rechou, A., Kirkwood, S., Arnault, J., and Dalin, P.: Short vertical-wavelength inertia-gravity waves generated by a jet-front system at Arctic latitudes - VHF radar, radiosondes and numerical modelling, Atmos. Chem. Phys., 14, 6785-6799, 10.5194/acp-14-6785-2014, 2014.

2) Figure 2 shows five existing thermal tropopause definitions, other contextual works that would help greatly in the presentation, framing, justification, and discussion of such work, which will help directly future efforts towards accomplishing this study or another iteration. Different definitions have their own advantages and limitations, so it is beneficial to understand the formation mechanisms of the tropopause and to further research stratosphere-troposphere exchange processes by defining the tropopause from various perspectives.

3) In the manuscript, in addition to LRT, we also compared it with CPT (see the Figure 10 in revised manuscript).

5. L239: could you explain what the Y-bar in the SST formula stands for. Is it the mean over the complete temperature profile or your defined altitude range?

**Reply:**

Thanks for your careful check. $\bar{Y}$ is the mean of the measurement temperature profiles.

6. Table 4: I am a bit confused about the presented numbers here and the

summary on DT given in L261. The table tells me 22.9% for BG and 17.5% for LRT DT-events. Why are the numbers in the text different?

**Reply:**

I apologize for my carelessness. Table 4 shows the correct results, and we have corrected the relevant text description.

7. Figure 8: Although you used a relative-TP coordinate system, I would expect some units and a y-range for Figure b to h. Otherwise you must add corresponding information into the caption.

**Reply:**

We added the scale value and units of the y axis for Figure 8 (b)–(h), and also added more explanation of the y axis in the figure caption.

8. Figure 9: this caption should include some more details on curves, symbols, color code and isolines (TP types).

**Reply:**

We've added descriptions in the figure caption. Figure 9: Height–time cross section of temperature profiles (shaded) from radiosondes in 2014 at Kuqa site (119.7 ºE, 49.25 ºN). DTH1 (magenta dotted line) and DTH2 (red dotted line) is defined the first and second tropopause height by bi-Gaussian, respectively. LRTH1 (black dotted line) and LRTH2 (blue dotted line) is defined the first and second tropopause height by LRT, respectively.

10. L438: 'In addiation, the statistical analysis …' please revise this sentence.

**Reply:**

We modified the sentence to "In addiation, the statistical analysis based on bi-Gaussian method are being analyzed in the future research work to get further understanding for tracer–tracer stratosphere–troposphere exchange, especially for the DT structures."

11. L490: 'Five-year (from 2012 to 2016) historical radiosondes' : 'Five-Year radiosonde data in China' might be better.

**Reply:**

Thank you for your comment. We have corrected it.

12. L494: why do the latitude belts overlap with each other? Should it be 25-35, which are also not really mid-high latitutdes?

**Reply:**

Thanks for your careful check. We have modified to "mid-high latitudes [25 ºN, 35 ºN]".

**Minor revision-egusphere-2024-345-referee-report-3**

**We appreciate the valuable review and constructive feedback provided by the reviewers. We agree with the reviewers' suggestions and carefully revise the manuscript. The detailed corrections are listed below.**

**Review of "A novel method to detect the tropopause structure based on bi-Gaussian function"**

The authors have put a solid amount of work into revising the manuscript to address my comments and those of the other reviewers. The clarity of the paper and its assertions have improved and many of my concerns have been reduced. However, there are still a handful of edits that I believe must be made before this can be considered for publication. Most are technical in nature, but there are two general points remaining that are most important in my view.

**General Comments**

1. The use of "local cold**est** point(s)" throughout is unnecessarily confusing. This would be much clearer stated as "local cold point(s)", since only one could possibly be the coldest.

**Reply:**

Thank you for your suggestion. We have revised "local coldest point(s)" to "local cold point(s)" throughout the manuscript.

2. Units for all equations and terms outlining the bi-Gaussian method are still absent from the paper. This includes Equations 5 & 6, where the resulting altitude units and dependent latitude units should be clearly defined, line 220 where the PTH height units for the search range should be defined, everything in Table 2, and everything in Table 3.

**Reply:**

Thank you for your suggestion. We have added the corresponding units to the locations where quantities appear for the first time in the revised manuscript.

3. Section 5 is somewhat vague, unclear, and includes a lot of hand-waving examples of double tropopause characteristics and purported linkages to other metrics. It is neither convincing or necessary based on my evaluation and I recommend simply removing it (lines 403-481; Figs. 13-15; Table 5) and related lines 503-507 from the manuscript.

**Reply:**

Thank you for your comment. We have deleted the discussion in Sec.5 and related expressions. At the same time, we will continue to improve this content in the next research work.

**Technical Edits**

**Reply:**

**Thanks for the reviewer's modification opinions on scientific writing, as below. We have made corresponding modifications and rewritten some unclear sentences.**

1. Line 12 – recommend revising second instance of "structures" to "characteristics.

2. Line 16 – "tropopauses" should be "tropopause"

3. Line 18 – "remarkable" should be deleted. This is subjective, non-scientific language.

4. Line 19 – delete "recognition"

5. Line 28 – "formatted" should be "formed"

6. Line 30 – recommend revising "the case of" to "with cases that" & delete "happens occasionally"

7. Line 37 – "tropopause" should be "tropopauses"

8. Line 43 – "of tropopause" should be "of the tropopause" & "formatted" should be deleted

9. Line 48 – "discovered the tropopause" should be "discovered tropopause"

10. Line 57 – "separated" should be "used"

11. Line 58 – "lapse minimum rate" should be "lapse rate minimum"

12. Line 65 – "called" should be "called the"

13. Line 71 – delete "tracers"

14. Line 75 – replace "compositions, with active STE" with "constituents"

15. Line 80 – "tropopause" should be "the tropopause"

16. Line 85 – "tropopause" should be "the tropopause"

17. Line 88 – "details" should be "detail" & "with the existing" should be "with existing"

18. Line 98 – delete "there are"

19. Line 147 – revise "influenced by stratospheric" to "constrained by"

20. Line 159 – "gradient" should be "gradients" & "compositions" should be "composition"

21. Line 164 – "following" should be "follows"

22. Line 169 – "following" should be "follows"

23. Line 179 – "LRT, provides" should be "LRT, and provides"

24. Line 197 – "minimum points" should be "minimum temperature points"

25. Line 202 – "following" should be "follows"

26. Line 252 – once again, "remarkable" should be deleted.

27. Line 261 – "general" should be "total" & the numbers here seem incorrect. If I total the lines from Table 4, I instead get 23.00% and 17.85%.

**Reply:**

I apologize for my carelessness. Table 4 shows the correct results, and we have corrected the relevant text description.

28. Line 289 – the dots are "magenta", not "cyan"

29. Line 309 – "contradictory" should be "contradiction"

30. Line 315 – insert "with a" before "prevailing ST structure"

31. Line 320 – start sentence with "The"

32. Line 322 – "in the Fig." should be "in Fig."

33. Line 323 – "bi-Gaussian" should be "the bi-Gaussian approach"

34. Line 335 – "if exist" should be "if multiple exist"

35. Line 343 – "accuracy" would be better stated as "agreement with the LRT"

36. Line 400 – "in UTLS" should be "in the UTLS"

37. Lines 409-411 – I do not know what this sentence means.

**Reply:**

This sentence, in the Sec.5, has been deleted.

38. Line 427 – what "invades"?

**Reply:**

This sentence, in the Sec.5, has been deleted.

39. Lines 428-429 – I also do not know what this sentence means.

**Reply:**

This sentence, in the Sec.5, has been deleted.

40. Line 440 – "at 315 K" should be "at the 315-K"

41. Lines 444-446 – this sentence does not begin correctly, but I'm not sure I know exactly how you want to relate it to something shown/mentioned previously.

**Reply:**

This sentence, in the Sec.5, has been deleted.

42. Line 459 – "does" should be "do"

43. Line 488 – "based bi-Gaussian" should be "based on bi-Gaussian identifications"

44. Line 491 – delete "monotonous"

Line 500 – delete "and uploading"